# Provably Efficient Neural Estimation of Structural Equation Model: An Adversarial Approach

**Luofeng Liao**
The University of Chicago
`luofengl@uchicago.edu`

**You-Lin Chen**
The University of Chicago
`youlinchen@uchicago.edu`

**Zhuoran Yang**
Princeton University
`zy6@princeton.edu`

**Bo Dai**
Google Research, Brain Team
`bodai@google.com`

**Zhaoran Wang**
Northwestern University
`zhaoranwang@gmail.com`

**Mladen Kolar**
The University of Chicago
`mkolar@chicagobooth.edu`

## Abstract

Structural equation models (SEMs) are widely used in sciences, ranging from economics to psychology, to uncover causal relationships underlying a complex system under consideration and estimate structural parameters of interest. We study estimation in a class of generalized SEMs where the object of interest is defined as the solution to a linear operator equation. We formulate the linear operator equation as a min-max game, where both players are parameterized by neural networks (NNs), and learn the parameters of these neural networks using the stochastic gradient descent. We consider both 2-layer and multi-layer NNs with ReLU activation functions and prove global convergence in an overparametrized regime, where the number of neurons is diverging. The results are established using techniques from online learning and local linearization of NNs, and improve in several aspects the current state-of-the-art. For the first time we provide a tractable estimation procedure for SEMs based on NNs with provable convergence and without the need for sample splitting.

## 1 Introduction

Structural equation models (SEMs) are widely used in economics [50], psychology [9], and causal inference [41]. In the most general form [41, 42], an SEM defines a joint distribution over $p$ observed random variables $\{X_j\}_{j=1}^p$ as $X_j = f_j(X_{\mathrm{pa}_D(j)}, \varepsilon_j)$, $j = 1, \ldots, p$, where $\{f_j\}$ are unknown functions of interest, $\{\varepsilon_j\}$ are mutually independent noise variables, $D$ is the underlying directed acyclic graph (DAG), and $\mathrm{pa}_D(j)$ denotes the set of parents of $X_j$ in $D$. The joint distribution of $\{X_j\}$ is Markov with respect to the graph $D$.

In most cases, estimation of SEMs are based on the conditional moment restrictions implied by the model. For example, some observational data can be thought of as coming from the equilibrium of a dynamic system. Examples include dynamic models where an agent interacts with the environment, such as in reinforcement learning [17], consumption-based asset pricing models [20], and rational expectation models [25]. In these models, the equilibrium behavior of the agent is characterized by conditional moment equations. A second example is instrument variable (IV) regression, where conditional moment equations also play a fundamental role. IV regression is used to estimate causal effects of input $X$ on output $Y$ in the presence of confounding noise $e$ [38]. Finally, in time-series and panel data models, observed variables exhibit temporal or cross-sectional dependence that can also be depicted by conditioning [46].

For these reasons, we study estimation of structural parameters based on the conditional moment restrictions implied by the model. We propose *the generalized structural equation model*, which takes the form of a linear operator equation

$$Af = b, \tag{1}$$

where $A : \mathcal{H} \to \mathcal{E}$ is a conditional expectation operator, which in most settings is only accessible by sampling, $\mathcal{H}$ and $\mathcal{E}$ are separable Hilbert spaces of square integrable functions with respect to some random variables, $f \in \mathcal{H}$ is the structural function of interest, and $b \in \mathcal{E}$ is known or can be estimated. Section 1.1 provides a number of important examples from causal inference and econometrics that fit into the framework (1).

Our contribution is threefold. **First**, we propose a new min-max game formulation for estimating $f$ in (1), where we parameterize both players by neural networks (NN). We derive a stochastic gradient descent algorithm to learn the parameters of both NNs. In contrast to several recent works that rely on RKHS theory [16, 36, 44], our method enjoys expressiveness thanks to the representation power of NNs. Moreover, our algorithm does not need sample splitting, which is a common issue in some recent works [26, 32]. **Second**, we analyze convergence rates of the proposed algorithm in the setting of 2-layer and deep NNs using techniques from online learning and neural network linearization. We show the algorithm finds a *globally optimal* solution as the number of iterations and the width of NNs go to infinity. In comparison, recent works incorporating NNs into SEM [26, 32, 7] lack convergence results. Furthermore, we derive a consistency result under suitable smoothness assumptions on the unknown function $f$. **Finally**, we demonstrate that our model enjoys wide application in econometric and causal inference literature through concrete examples, including non-parametric instrumental variable (IV) regression, supply and demand equilibrium model, and dynamic panel data model.

## 1.1 Examples of generalized SEM

We describe three examples of generalized SEM: IV regression, simultaneous equations models, and dynamic panel data model. In Appendix A, we introduce two more examples: proxy variables of unmeasured confounders in causal inference [34] and Euler equations in consumption-based asset pricing model [20]. Other examples that fit into the generalized SEM framework, but are not detailed in the paper, include nonlinear rational expectation models [25], policy evaluation in reinforcement learning, inverse reinforcement learning [40], optimal control in linearly-solvable MDP [16], and hitting time of stationary process [16].

**Example 1** (Instrumental Variable Regression [38, 26, 28]). In many applied problems endogeneity in regressors arises from omitted variables, measurement error, and simultaneity [50]. IV regression provides a general solution to the problem of endogenous explanatory variables. Without loss of generality, consider the model of the form

$$Y = g_0(X) + \varepsilon, \quad \mathbb{E}[\varepsilon \mid Z] = 0, \tag{2}$$

where $g_0$ is the unknown function of interest, $Y$ is the response, $X$ is a vector of explanatory variables, $Z$ is a vector of instrument variables, and $\varepsilon$ is the noise term. To see how the model fits our framework, define the operator $A : L^2(X) \to L^2(Z)$, $(Ag)(z) = \mathbb{E}[g(X) \mid Z = z]$. Let $b(z) = \mathbb{E}[Y \mid Z = z] \in L^2(Z)$. The structural equation (2) can be written as $Ag = b$.

**Example 2** (Simultaneous Equations Models). Dynamic models of agent's optimization problems or of interactions among agents often exhibit simultaneity. Consider a demand and supply model as a prototypical example [33]. Let $Q$ and $P$ denote the quantity sold and price of a product, respectively. Then

$$\begin{aligned} Q = D(P, I) + U_1, \ P = S(Q, W) + U_2, \\ \mathbb{E}[U_1 \mid I, W] = 0, \ \mathbb{E}[U_2 \mid I, W] = 0, \end{aligned} \tag{3}$$

where $D$ and $S$ are functions of interest, $I$ denotes consumers' income, $W$ denotes producers' input prices, $U_1$ denotes an unobservable demand shock, and $U_2$ denotes an unobservable supply shock. Each observation of $\{P, Q, I, W\}$ is a solution to the equation (3). In Appendix A we cast it into the form (1). The knowledge of $D$ is essential in predicting the effect of financial policy. For example, let $\tau$ be a percentage tax paid by the purchaser. Then the resulting equilibrium quantity is the solution $\hat{Q}$ to the equation $\hat{Q} = D\big((1 + \tau)(S(\hat{Q}, I) + U_1), W\big) + U_2$.

**Example 3** (Dynamic Panel Data Models [46]). Exploiting how outcomes vary across units and over time in the dataset is a common approach to identifying causal effects [1]. Panel data are comprised

of observations of multiple units measured over multiple time periods. We consider a dynamic model that includes time-varying regressors and allows us to investigate the long-run relationship between economic factors [46]:

$$
\begin{aligned}
Y_{it} &= m\left(Y_{i,t-1}, X_{it}\right) + \alpha_i + \varepsilon_{it}, \\
\mathbb{E}[\varepsilon_{it} \mid \underline{Y}_{i,t-1}, \underline{X}_{it}] &= 0, \quad i = 1, \ldots, N, \quad t = 1, \ldots, T.
\end{aligned}
\tag{4}
$$

Here $X_{it}$ is a $p \times 1$ vector of regressors, $m$ is the unknown function of interest, $\alpha_i$'s are the unobserved individual-specific fixed effects, potentially correlated with $X_{it}$, and $\varepsilon_{it}$'s are idiosyncratic errors. $\underline{X}_{it} := (X_{it}^\top, \ldots, X_{i1}^\top)^\top$ and $\underline{Y}_{i,t-1} := (Y_{i,t-1}, \ldots, Y_{i1})^\top$ are the history of individual $i$ up to time $t$. After first differencing, we can cast (4) into equation of the form (1) (see Appendix A).

## 1.2 Related work

**Neural networks in structural equation models**. IV regression and generalized method of moments (GMM) [24] are two important tools in structural estimation. For example, the work of Blundell et al. [8] estimates system of nonparametric demand curves with endogeneity and a sieve-based measure of ill-posedness of the statistical inverse problem is introduced. The work of Chen and Pouzo [14] allows for various convex or/and lower-semicompact penalization on unknown structural functions. Typical nonparametric approaches to to IV regression include kernel density estimators [38, 12] and spline regression [18, 11]. However, traditional nonparametric methods usually suffer from the curse of dimensionality and the lack of guidance on the choice of kernels and splines.

Existing work on structural estimation using NNs, best to our knowledge, includes Deep IV [26], Deep GMM [7] and Adversarial GMM [32]. However, due to the artifacts in saddle-point problem derivation and non-linearity of NNs, these methods suffer from computational cost [26], the need of sample splitting [32, 26] or lack of convergence results [7, 32]. The use of NN also appears in works in econometrics. The work of Chen and Ludvigson [13] applies NN to estimate unknown habit function in consumption based asset pricing model. The work of Farrell et al. [21] discusses the use of NN in semi-parametric estimation but not computational issues.

Kernel IV [44] and Dual IV [36] apply reproducing kernel Hilbert space (RKHS) theory to IV regression. Dual IV is closely related to the work of Dai et al. [16], where the authors discuss problems of the form $\min_f \mathbb{E}_{x,y}[\ell(y, \mathbb{E}_{z|x}[f(x,z)])]$ and reformulate it as a min-max problem using duality, interchangeability principle, and dual continuity. In Appendix F, we show that our minimax formulation of IV has a natural connection to GMM compared to Dual IV.

Finally, we notice an excellent concurrent work [19] which discusses the statistical property of a class of minimax estimator for conditional moment restriction problems. The proposed estimator in that paper is almost identical to ours, and yet we focus on showing convergence of training with NN using neural tangent kernel theory. There is significant distinction from their work.

**Neural tangent kernel and overparametrized NN**. Recent work on neural tangent kernel (NTK) [29] shows that in the limit when the number of neurons goes to infinity, the nonlinear NN function can be represented by a linear function specified by the NTK. Consequently, the optimization problem parametrized by NNs reduces to a convex problem, and can be tackled by tools in classical convex optimization. Examples following this idea include [10, 48, 51]. In fact, the present paper follows a similar philosophy, by reducing the analysis of neural gradient update to regret analysis of convex online learning, in the presence of bias and noise in the gradient. Finally, the present work is also related to recent advances in overparametrized NNs [3, 2, 22, 29, 31, 52, 39]. These works point out that NNs exhibit an implicit local linearization which allows us to interpret the former as a linear function when they are trained using gradient type methods. The present paper is built on an adaptation of these results.

## 1.3 Notations

We call $(f^*, u^*) \in \mathcal{F} \times \mathcal{U}$ a saddle point of a function $\phi : \mathcal{F} \times \mathcal{U} \to \mathbb{R}$ if for all $f \in \mathcal{F}$, $u \in \mathcal{U}$, $\phi(f^*, u) \leqslant \phi(f^*, u^*) \leqslant \phi(f, u^*)$. The indicator function $\mathbb{1}\{\cdot\}$ is defined as $\mathbb{1}\{A\} = 1$ if the event $A$ is true; otherwise $\mathbb{1}\{A\} = 0$. Let $[n] = \{1, 2, \ldots, n\}$. For two sequences $\{a_n\}, \{b_n\}$, the notation $b_n = \mathcal{O}(a_n)$ represents that there exists a constant $C$ such that $b_n \leqslant Ca_n$ for all large $n$. We write $a_n \sim b_n$ if $a_n = O(b_n)$ and $b_n = O(a_n)$. The notation $\tilde{\mathcal{O}}$ ignores logarithmic factors. For a matrix $A$, let $\|A\|_F$ be the Frobenius norm.

For a probability space $(\Omega, \mathcal{F}, \mathbb{P})$, let $X : \Omega \to \mathbb{R}^p$ be a $p$-dimension random vector. The probability distribution of $X$ is characterized by its joint cumulative distribution function $F$. Partition $X$ into $X = [X_1^\top, X_2^\top]^\top$ where $X_1 \in \mathbb{R}^{p_1}, X_2 \in \mathbb{R}^{p_2}$, and let $F_{X_1}, F_{X_2}$ be the marginal distribution functions, respectively. Denote by $L_F^2(\mathbb{R}^{p_1}, F_{X_1}) = \{f_1 : \mathbb{R}^{p_1} \to \mathbb{R} : \mathbb{E}_{X_1}[f_1(X_1)^2] < \infty\}$ the Hilbert space of real-valued square integrable functions of $X_1$ and similarly define $L_F^2(\mathbb{R}^{p_2}, F_{X_2})$. For ease of presentation we denote $L_F^2(\mathbb{R}^{p_1}, F_{X_1})$ by $L^2(X_1)$ when the context is clear. For $f, g \in L_F^2$, the inner product is defined by $\langle f, g \rangle_{L^2(X)} = \mathbb{E}_X[f(X)g(X)]$. For a linear operator $A : \mathcal{H} \to \mathcal{E}$ denote by $\mathcal{N}(A) = \{f \in \mathcal{H} : Af = 0\}$ its null space. Denote by $A^*$ the adjoint of a bounded linear operator $A$. For a subspace $B \subset \mathcal{H}$ in a Hilbert space $\mathcal{H}$, denote by $B^\perp = \{a \in \mathcal{H} : \langle a, b \rangle_{\mathcal{H}} = 0, \forall b \in B\}$ the orthogonal complement of $B$ in $\mathcal{H}$.

## 2 Adversarial SEM

We formalize our problem setup and introduce the Tikhonov regularized method for finding a solution for the operator equation in (1) in Section 2.1. In Section 2.2 we derive a saddle-point formulation of our problem. The players of the resulting min-max game are parametrized by NNs, detailed in Section 3.

### 2.1 Problem setup

Let $X = [X_1^\top, X_2^\top]^\top$ be a random vector with distribution $F_X$. Let $F_{X_1}, F_{X_2}$ be the marginal distributions of $X_1$ and $X_2$, respectively. We assume there are no common elements in $X_1$ and $X_2$. Furthermore, suppose there is a regular conditional distribution for $X_1$ given $X_2$. Let $\mathcal{H} = L^2(X_1)$ and $\mathcal{E} = L^2(X_2)$. We let $A : \mathcal{H} \to \mathcal{E}$ be the conditional expectation operator defined as

$$(Af)(\cdot) = \mathbb{E}[f(X_1) \mid X_2 = \cdot].$$

We want to estimate the solution $f$ to the equation (1), $Af = b$, for some known or estimable $b \in \mathcal{E}$. In statistical learning literature, (1) is called *stochastic ill-posed problem* when $b$ or both $A$ and $b$ have to be estimated [47]. In the linear integral equation literature, when $A$ is compact, (1) belongs to the class of Fredholm equations of type I. An inverse problem perspective on conditional moment problems is provided in [11].

A compact operator[1] with infinite dimensional range cannot have a continuous inverse [11], which raises concerns about stability of operator equation (1). A classical way to overcome the problem of instability is to look for a Tikhonov regularized solution, which is uniquely defined [30]. For all $b \in \mathcal{E}, \alpha > 0$, we define the Tikhonov regularized problem

$$f^\alpha = \arg\min_{f \in \mathcal{H}} \tfrac{1}{2}\|Af - b\|_{\mathcal{E}}^2 + \tfrac{\alpha}{2}\|f\|_{\mathcal{H}}^2. \qquad (5)$$

### 2.2 Saddle-point formulation

From an optimization perspective, problem (5) is difficult to solve in that the conditional expectation operator is nested inside the square loss.

First, it is difficult to estimate the conditional expectation in some cases since we have only limited samples coming from the conditional distribution $p(X_1 \mid X_2)$. In the extreme case, for each value of $X_2 = x_2$ we only observe one sample.

Second, when approximating $f$ using some parametrized function class, we encounter the so-called double-sample issue. Let's investigate Example 1. In the nonparametric IV problem $\mathbb{E}[g(X)|Z] = \mathbb{E}[Y|Z]$, we want to estimate $g$. Consider the square loss $L(g) = \mathbb{E}_Z\big[\big(\mathbb{E}[g(X) - Y|Z]\big)^2\big]$. Assume $g$ is approximated by a function parameterized with $\theta$. Taking the gradient w.r.t. $\theta$ and assuming exchange of $\nabla_\theta$ and $\mathbb{E}$, we get $\nabla_\theta L(g) = 2\mathbb{E}_Z\big[\mathbb{E}[g(X) - Y|Z] \cdot \mathbb{E}[\nabla_\theta g(X)|Z]\big]$. Assume we observe iid samples of $(X, Y, Z)$. The product of two conditional expectation terms implies that, to obtain an unbiased estimate of the gradient, we will need two samples of $(X, Y, Z)$ with $Z$ taking the same value. This is usually unlikely except for simulated environments.

Our saddle-point formulation circumvents such problems by using the probabilistic property of conditional expectation. The proposed method also offers some new insights into the saddle-point formulation of IV regression [36, 7], which shows our derivation is closely related to GMM. This is discussed in Appendix F.

Now we derive a min-max game formulation for (5). Assume $b$ is known. Let $R : \mathcal{H} \to \mathbb{R}_+$ be some suitable norm on $\mathcal{H}$ that captures smoothness of the function $f$. We consider the constrained form of minimization problem (5): $\min_{f \in \mathcal{H}} \frac{1}{2} R(f)$ subject to $Af = b$. For some positive number $\alpha > 0$, we define the Lagrangian with penalty on the multiplier $u \in \mathcal{E}$,

$$\tilde{L}(f, u) = \tfrac{1}{2} R(f) + \langle Af - b, u \rangle_\mathcal{E} - \tfrac{\alpha}{2} \|u\|_\mathcal{E}^2.$$

Without loss of generality, we move the penalty level $\alpha$ to $R(f)$. Finally, using a property of conditional expectation that $\langle Af, u \rangle_\mathcal{E} = \mathbb{E}_{X_2}\big[\mathbb{E}[f(X_1) \mid X_2] u(X_2)\big] = \mathbb{E}[f(X_1) u(X_2)]$ and choosing $R(f) = \|f\|_\mathcal{H}^2$, we arrive at our saddle-point problem

$$\min_{f \in L^2(X_1)} \max_{u \in L^2(X_2)} \mathbb{E}\big[\big(f(X_1) - b(X_2)\big) u(X_2) + \tfrac{\alpha}{2} f(X_1)^2 - \tfrac{1}{2} u(X_2)^2\big]. \tag{6}$$

We remark that as long as $R(f)$ can be estimated from samples, our subsequent algorithm and analysis work with some adaptations. Note that the above derivation is also suitable for equations of the form $(I - K)f = b$, where $K$ is a conditional expectation operator (e.g., Example 4 in Appendix A). Moreover, the function $b$ can be either known or estimable from the data, i.e., $b$ can be of the form $b(X_2) = \mathbb{E}[\tilde{b}(X_1, X_2) \mid X_2]$ where $\tilde{b}$ is known.

# 3 Neural Network Parametrization

The recent surge of research on the representation power of NNs [29, 3, 2, 31, 4, 10] motivates us to use NNs as approximators in (6). Consider the 2-layer NN with parameters $B$ and $m$ (to be defined in (9)). As the width of the NN, $m$, goes to infinity, the class NNs approximates a subset of the reproducing kernel Hilbert space induced by the kernel $K(x, y) = \mathbb{E}_{a \sim \mathcal{N}(0, \frac{1}{d} I_d)}\big[\mathbb{1}\left\{a^\top x > 0\right\} \mathbb{1}\left\{a^\top y > 0\right\} x^\top y\big]$. Such a subset is a ball with radius $B$ in the corresponding RKHS norm. This function class is sufficiently rich, if the width $m$ and the radius $B$ are sufficiently large [4].

However, due to non-linearity of NNs, to devise an algorithm for the NN-parametrized problem (6) with theoretical guarantee is no easy task. In this section, we describe the NN parametrization scheme and the algorithm. As a main contribution of the paper, we then provide formal statements of results on convergence rate and estimation consistency.

To keep the notation simple, we assume $X_1$ and $X_2$ are of the same dimension $d$. We parametrize the function spaces $L^2(X_1)$ and $L^2(X_2)$ in (6) by a space of NNs, $\mathcal{F}_{\mathrm{NN}}$, defined in (9) and (10) below. With this parameterization, we write the primal problem in (5) as

$$\min_{f \in \mathcal{F}_{\mathrm{NN}}} L(f) := \tfrac{1}{2} \|Af - b\|_\mathcal{E}^2 + \tfrac{\alpha}{2} \|f\|_\mathcal{H}^2, \tag{7}$$

and the min-max problem in (6) becomes

$$\min_{f \in \mathcal{F}_{\mathrm{NN}}} \max_{u \in \mathcal{F}_{\mathrm{NN}}} \phi(f, u) := \mathbb{E}\big[\big(f(X_1) - b(X_2)\big) u(X_2) + \tfrac{\alpha}{2} f(X_1)^2 - \tfrac{1}{2} u(X_2)^2\big]. \tag{8}$$

Problem (8) involves simultaneous optimization over two NNs. Notice that for each fixed $f$ in the outer minimization, the maximum of the inner maximization over $\mathcal{E}$ is attained at $u(\cdot) = \mathbb{E}[f(X_1) \mid X_2 = \cdot] - b(\cdot) \in \mathcal{E}$. This can be seen by noting $\max_{u \in \mathcal{E}} \{\langle Af - b, u \rangle_\mathcal{E} - \frac{1}{2} \|u\|_\mathcal{E}^2\} = \frac{1}{2} \|Af - b\|_\mathcal{E}^2$. If for all $f \in \mathcal{F}_{\mathrm{NN}}$ such maximum is attained in $\mathcal{F}_{\mathrm{NN}}$, then every primal solution $f^* \in \mathcal{F}_{\mathrm{NN}}$ in the saddle point of (8), $(f^*, u^*)$, is also an optimal solution to the problem (7).

Next we introduce the function classes of NNs and the initialization schemes.

## 3.1 Neural Network Parametrization

**2-layer NNs.** Consider the space of 2-layer NNs with ReLU activations and initialization $\Xi_0 = [W(0), b_1, \ldots, b_m]$

$$\mathcal{F}_{d,B,m}(\Xi_0) = \left\{ x \mapsto \frac{1}{\sqrt{m}} \sum_{r=1}^{m} b_r \sigma(W_r^\top x) : W \in S_B \right\}, \tag{9}$$

where $\sigma(z) = \mathbb{1}\{z > 0\} \cdot z$ is the ReLU activation, $b_1, \ldots, b_m$ are scalars, and

$$S_B = \left\{ W \in \mathbb{R}^{md} : \|W - W(0)\|_2 \leqslant B \right\}$$

is the $B$-sphere centered at the initial point $W(0) \in \mathbb{R}^{md}$. Here we denote succinctly by $W$ the weights of a 2-layer NN stacked into a long vector of dimension $md$, and use $W_r \in \mathbb{R}^m$ to access the weights connecting to the $r$-th neuron, i.e., $W = [W_1^\top, \ldots, W_m^\top]^\top$. Each function in $\mathcal{F}_{B,m}$ is differentiable with respect to $W$, 1-Lipschitz, and bounded by $B$. We state the following distributional assumption on initialization.

**Assumption A.1** (NN initialization, 2-layer, [29])**.** *Consider the 2-layer NN function space $\mathcal{F}_{d,B,m}(\Xi_0)$ defined in (9). All initial weights and parameters, collected as $\Xi_0 = [W(0), b_1, \cdots, b_m]$, are independent, and generated as $b_r \sim \text{Uniform}(\{-1, 1\})$ and $W(0) \sim \mathcal{N}\left(0, \frac{1}{d}\mathbf{I}_d\right)$. During training we fix $\{b_r\}_{r=1}^m$ and update $W$.*

**Multi-layer NNs.** The class of $H$-layer NN, $\mathcal{F}_{d,B,H,m}$ with initialization $\Xi_{H,0} = \left\{A, \{W^{(h)}(0)\}_{h=1}^H, b\right\}$ is defined as

$$\mathcal{F}_{d,B,H,m}(\Xi_{H,0}) = \left\{ x \mapsto b^\top x^{(H)} \text{ where } x^{(h)} = \frac{1}{\sqrt{m}} \cdot \sigma(W^{(h)} x^{(h-1)}), h \in [H], \right.$$
$$\left. x^{(0)} = Ax : W \in S_{B,H} \right\}, \quad (10)$$

where $W = (\text{vec}(W^{(1)})^\top, \ldots, \text{vec}(W^{(H)})^\top)^\top \in \mathbb{R}^{Hm^2}$ is the collection of weights $W^{(h)} \in \mathbb{R}^{m^2}$ from all middle layers, $x^{(h)}$ is the output from the $h$-th layer, $A \in \mathbb{R}^{m \times d}$, $b \in \mathbb{R}^m$, the function $\sigma$ is applied element-wise, and

$$S_{B,H} = \left\{ W \in \mathbb{R}^{Hm^2} : \|W^{(h)} - W^{(h)}(0)\|_{\mathrm{F}} \leqslant B \text{ for any } h \in [H] \right\}.$$

We use the following initialization scheme.

**Assumption A.2** (NN initialization, multi-layer, [3, 22])**.** *Consider the space of multi-layer NNs $\mathcal{F}_{d,B,H,m}(\Xi_{H,0})$ defined in (10). Each entry of $A$ and $\{W^{(h)}(0)\}_{h=1}^H$ is independently initialized by $N(0,2)$, and entries of $b$ are independently initialized by $N(0,1)$. Assume $m = \Omega(d^{3/2} B^{-1} H^{-3/2} \log^{3/2}(m^{1/2} B^{-1}))$ and $B = \mathcal{O}(m^{1/2} H^{-6} \log^{-3} m)$. All initial parameters, $\Xi_{H,0} = \left\{A, \{W^{(h)}(0)\}_{h=1}^H, b\right\}$, are independent. During training we keep $A, b$ fixed and update $W$.*

We overload notations and denote by $\mathbb{E}_{\text{init}}[\cdot]$ the expectation taken over the random variables $\Xi_0$ or $\Xi_{H,0}$, the randomness of NN initialization.

## 3.2 Algorithm

To describe an implementable algorithm, we rewrite the saddle-point problem (8) in terms of NN weights. Denote the weights of NNs $f$ and $u$ in (8) by $\theta$ and $\omega$, respectively. Now $\theta$ and $\omega$ play the role of $W$ in (9) (resp. (10)) since during training only the weights $W$ in (9) (resp. (10)) are updated. For brevity set $f_\theta(\cdot) = f(\theta; \cdot)$ and $u_\omega(\cdot) = u(\omega; \cdot)$. With a slight abuse of notation, we use $\phi(\theta, \omega)$ and $\phi(f_\theta, u_\omega)$ (defined in (8)) interchangeably. Note $\phi$ is convex in the NN $f_\theta$ but not in the NN weights $\theta$. Let $F(\theta, \omega; x_1, x_2) = u_\omega(x_2) f_\theta(x_1) - u_\omega(x_2) b(x_2) - \frac{1}{2} u_\omega^2(x_2) + \frac{\alpha}{2} f_\theta^2(x_1)$. The saddle-point problem (8) is now rewritten as

$$\min_{\theta \in S_B} \max_{\omega \in S_B} \phi(\theta, \omega) = \mathbb{E}\left[ F(\theta, \omega; X_1, X_2) \right]. \quad (11)$$

Algorithm 1 is the proposed stochastic primal-dual algorithm for solving the game (8). Given initial weights $\theta_1$ and $\omega_1$, stepsize $\eta$, and i.i.d. samples $\{X_{1,t}, X_{2,t}\}$, for $t = 2, \ldots, T-1$,

$$\theta_{t+1} = \Pi_{S_B}\left(\theta_t - \eta \nabla_\theta F(\theta_t, \omega_t; X_{1,t}, X_{2,t})\right),$$
$$\omega_{t+1} = \Pi_{S_B}\left(\omega_t + \eta \nabla_\omega F(\theta_t, \omega_t; X_{1,t}, X_{2,t})\right). \quad \text{(Algorithm 1)}$$

Here $\Pi_{S_B}$ is the projection operator. The search spaces in (11) and the projection operator should be replaced by $S_{B,m}$ and $\Pi_{S_{B,m}}$, respectively, when multi-layer NNs are used. If $b$ takes the form $b(X_2) = \mathbb{E}[\tilde{b}(X_1, X_2) \mid X_2]$ our algorithm proceeds by replacing $b$ in $F$ with $\tilde{b}$.

Define by $\mathcal{D} = \sigma\big\{\{X_{1,t}, X_{2,t}\}_{t=1}^T\big\}$ the $\sigma$-algebra generated by the training data. Define the average of NNs as our final output

$$\bar{f}_T(\cdot) = \frac{1}{T}\sum_{t=1}^T f(\theta_t; \cdot). \tag{12}$$

## 4 Main results

Due to nonlinearity of NNs, $\phi$ is not convex-concave in $(\theta, \omega)$, which makes the analysis of Algorithm 1 difficult. However, as will be shown in Theorem 4.1, under certain assumptions Algorithm 1 enjoys *global convergence* as $T$ and $m$ go to infinity. For a candidate solution $f$, we consider the suboptimality $E(f)$ as a measure of quality of solution, i.e.,

$$E(f) = L(f) - L^*, \tag{13}$$

where $L^* = \min_{f \in \mathcal{F}_{B,m}} L(f)$ is the minimum value of $L$ over the space of NNs. We define similar quantities when multi-layer NNs are used. Next we describe regularity assumptions on the data distribution.

**Assumption A.3** (Bounded support and bounded range). *Assume* $\max\{\|X_1\|_2, \|X_2\|_2\} \leqslant 1$ *almost surely. Assume* $b(X_2)$ *is bounded almost surely.*

**Assumption A.4** (Regularity of data distribution). *Assume that there exists* $c > 0$, *such that for any unit vector* $v \in \mathbb{R}^p$ *and any* $\zeta > 0$, $\mathbb{P}(|v^\top X_1| \leqslant \zeta) \leqslant c\zeta$, $\mathbb{P}(|v^\top X_2| \leqslant \zeta) \leqslant c\zeta$.

**Assumption A.5** (The conditional expectation operator is closed in $\mathcal{F}_{\mathrm{NN}}$). *With high probability with respect to NN initialization, for any* $f \in \mathcal{F}_{B,m}$ *(or* $\mathcal{F}_{B,H,m}$*),* $u(\cdot) = \mathbb{E}[f(X_1) \mid X_2 = \cdot] - b(\cdot)$ *belongs to the class* $\mathcal{F}_{B,m}$ *(or* $\mathcal{F}_{B,H,m}$*).*

In Assumption A.3, boundedness of the random variable $b(X_2)$ is satisfied in common applications. Assumption A.4 is used when invoking the linearization effects of 2-layer NNs; see Lemma 5.1. Assumption A.5 ensures the connection between the min-max problem (8) and the primal problem (7). Assumption A.5 can be removed by incorporating an approximation error term in the error bound. We are ready to state the global convergence results for Algorithm 1. The proof is presented in Appendix D.7.

**Theorem 4.1** (Global convergence of Algorithm 1). *Consider the iterates generated by Algorithm 1 with stepsize* $\eta$. *Let* $a = \max\{\alpha, 1\}$. *Recall* $\mathbb{E}_{\mathrm{init}}[\cdot]$ *is the expectation w.r.t. NN initialization. For the averaged NN* $\bar{f}_T$ *(defined in (12)), its suboptimality* $E(\bar{f}_T)$ *(defined in (13)) satisfies the following bounds.*

*1. (2-layer NNs) Under Assumption A.1, A.3, A.4 and A.5, with probability over* $1 - 2\delta$ *with respect to the training data* $\mathcal{D}$,

$$\mathbb{E}_{\mathrm{init}}\big[E(\bar{f}_T)\big] = \mathcal{O}\Big(a\eta B + \frac{B}{T\eta} + \frac{aB^{3/2}\log^{1/2}(1/\delta)}{T^{1/2}} + \frac{aB^{5/2}}{m^{1/4}}\Big). \tag{14}$$

*2. (Multi-layer NNs) Under Assumption A.2, A.3 and A.5, with probability over* $1 - c\delta - c\exp\big(\Omega(\log^2 m)\big)$ *with respect to the training data* $\mathcal{D}$ *and NN initialization* $\Xi_{H,0}$,

$$E(\bar{f}_T) = \mathcal{O}\Big(P_1\eta a\log m + \frac{P_2}{T\eta} + \frac{P_3 a\log m \log^{1/2}(1/\delta)}{T^{1/2}} + \frac{P_4 a\log^{3/2} m}{m^{1/6}}\Big),$$

*where* $P_1 = H^4 B^{4/3}$, $P_2 = H^{1/2}B$, $P_3 = H^5 B^2$, $P_4 = H^6 B^3$, *and* $c$ *is an absolute constant.*

Each of the error rates in Theorem 4.1 consists of two parts: the optimization error and the linear approximation error; see Section 5.2 for a detailed derivation. For the two-layer case, if the total training step $T$ is known in advance, the optimal stepsize choice is $\eta \sim T^{-1/2}$, and the resulting error rate is $\tilde{\mathcal{O}}(T^{-1/2} + m^{-1/4})$. The optimization error term $\mathcal{O}(T^{-1/2})$ is comparable to the rate in [37] where stochastic mirror descent method is used in stochastic saddle-point problems. Importantly, the error bound (14) converges to zero as $T, m \to \infty$. For the multi-layer case, optimizing $\eta$ yields the error rate $\tilde{\mathcal{O}}(T^{-1/2} + m^{-1/6})$; the linear approximation error has increased due to the highly non-linear nature of multi-layer NNs.

## 4.1 Consistency

If we assume smoothness of the solution $f$ to the operator equation $Af = b$ defined in (1) and compactness of the operator $A$, we are able to control the rate of regularization bias. For a compact operator $A$, let $\{\lambda_j, \phi_j, \psi_j\}_{j=1}^\infty$ be its singular system [30], i.e., $\{\phi_j\}$ and $\{\psi_j\}$ are orthonormal sequences in $\mathcal{H}, \mathcal{E}$, repectively, $\lambda_j \geqslant 0$, and satisfy $A\phi_j = \lambda_j\psi_j$, $A^*\psi_j = \lambda_j\phi_j$, where $A^*$ is the adjoint operator of $A$. For any $\beta > 0$, define the $\beta$-regularity space [11]

$$\Phi_\beta = \left\{ f \in \mathcal{N}(A)^\perp \text{ such that } \sum_{j=1}^\infty \frac{\langle f, \phi_j \rangle_\mathcal{H}^2}{\lambda_j^{2\beta}} < \infty \right\} \subset \mathcal{H}. \tag{15}$$

Equipped with the definition of $\beta$-regularity space, we are now ready to state the consistency result for 2-layer NN. The proof is presented in Appendix D.8.

**Assumption A.6** (Zero approximation error). *The primal problems* (5) *and* (7) *yield the same solution.*

**Assumption A.7** (Smoothness of the truth). *Assume the operator $A$ defined in* (1) *is injective and compact, and that $f$, the unique solution to* (1)*, lies in the regularity space $\Phi_\beta$ defined in* (15) *for some $\beta > 0$.*

**Theorem 4.2** (Consistency, 2-layer NN). *Consider the iterates generated by Algorithm 1 with stepsize $\eta \sim (aT)^{-1/2}$, where $a = \max\{\alpha, 1\}$. Assume A.1, A.3, A.4, A.5, A.6 and A.7. Then with probability at least $1 - \delta$ over the sampling process,*

$$\mathbb{E}_{\text{init}}\left[ \|\bar{f}_T - f\|_{L^2(X_1)}^2 \right] = C\left( \alpha^{\min\{\beta, 2\}} + \frac{1}{\alpha\sqrt{a}} \frac{1}{T^{1/2}} + \frac{a}{\alpha}\left( \frac{1}{T^{1/2}} + \frac{1}{m^{1/4}} \right) \right), \tag{16}$$

*where $\bar{f}_T$ is defined in* (12)*, $f$ in Assumption A.7, and $C$ is a constant independent of $\beta, \alpha, T$ and $m$.*

If $0 < \beta \leqslant 2$ and $0 < \alpha \leqslant 1$, the optimal choice of $\alpha$ is $\alpha \sim (T^{-1/2} + m^{-1/4})^{1/(\beta+1)}$, assuming $T$ and $m$ are large enough, and the estimation error (16) is of order $\mathcal{O}\left( (T^{-1/2} + m^{-1/4})^{\beta/(\beta+1)} \right)$. To the best of our knowledge, this is the first estimation error rate of structural equation models using NNs. We remark [21] also provides bounds on the estimation error of an NN-based estimator in the setting of semi-parametric inference, but they do not discuss computational issues.

# 5 Proof sketch

## 5.1 Local linearization of NNs

The key observation is that as the width of NN increases, NN exhibits similar behavior to its linearized version [2]. For an NN $f \in \mathcal{F}_{B,m}$ (or $\mathcal{F}_{B,H,m}$, with slight notation overload), we denote its linearized version at $W(0)$ by

$$\widehat{f}(x, W) = f(x, W(0)) + \left\langle \nabla_W f\big(x, W(0)\big), W - W(0) \right\rangle. \tag{17}$$

The following lemma offers a precise characterization of linearization error for 2-layer NNs; the proof is presented in Section D.1 in Appendix D. Essentially, it shows that for 2-layer NNs the expected approximation error of the function $f(\cdot, W)$ by $\widehat{f}(\cdot, W)$ decays at the rate $\mathcal{O}(m^{-1/2})$, for any $W \in S_B$. In other words, as the width of NN goes to infinity, the NN function behaves like a *linear function*. Similar results on approximation error for multi-layer NNs hold; see Appendix B.

**Lemma 5.1** (Error of local linearization, 2-layer). *Consider the 2-layer neural networks in* (9)*. Assume that there exists $c > 0$, for any unit vector $v \in \mathbb{R}^d$ and any constant $\zeta > 0$, such that $\mathbb{P}_X(|v^\top X| \leqslant \zeta) \leqslant c\zeta$. Under Assumption A.1 we have for all $W \in S_B$ and all $x$,*

$$\mathbb{E}_{\text{init}, X}\left[ |f(X, W) - \widehat{f}(X, W)|^2 \right] = \mathcal{O}(B^3 m^{-1/2}), \quad and$$

$$\mathbb{E}_{\text{init}, X}\left[ \|\nabla_W f(X, W) - \nabla_W \widehat{f}(X, W)\|^2 \right] = \mathcal{O}(B m^{-1/2}).$$

## 5.2 Convergence analysis

In this section, we discuss techniques used to bound the minimization error via the analysis of the regret, in the case of 2-layer NNs. The same reasoning applies to the maximizing player $\omega$ and extension to multi-layer NN is obvious. The following lemma relates regret and primal error. The proof is presented in Appendix D.3.

**Lemma 5.2** (A bound on primal error). *Consider a sequence of candidates $\{(f_t, u_t)\}_{t=1}^T$ for the minimax problem* (8) *that satisfy the following regret bounds*

$$\frac{1}{T}\sum_{t=1}^T \phi(f_t, u_t) \leqslant \min_{f \in \mathcal{F}_{NN}} \frac{1}{T}\sum_{t=1}^T \phi(f, u_t) + \epsilon_f, \quad \frac{1}{T}\sum_{t=1}^T \phi(f_t, u_t) \geqslant \max_{u \in \mathcal{F}_{NN}} \frac{1}{T}\sum_{t=1}^T \phi(f_t, u) - \epsilon_u.$$

(18)

*Denote $\bar{f}_T = \frac{1}{T}\sum_{t=1}^T f_t$. If Assumption A.5 holds, then $E(\bar{f}_T) = L(\bar{f}_T) - L^* \leqslant \epsilon_f + \epsilon_u$.*

The above lemma suggests we separate our analysis for the two players. For example, to analyze $\epsilon_f$ we can think of the sequence $\{u_t\}$ as fixed and find an upper bound of the quantity $\frac{1}{T}\sum_{t=1}^T \phi(f_t, u_t) - \frac{1}{T}\sum_{t=1}^T \phi(f, u_t)$. We will demonstrate our proof idea via the analysis of $\epsilon_f$; it can easily extend to $\epsilon_u$.

We focus on the analysis of the minimizer $\theta$ and therefore we denote $\phi_t(\cdot) = \phi(\cdot, \omega_t)$ (defined in (8)). Also let $\widehat{\phi}_t(\theta) = \mathbb{E}_X[\widehat{u}_{\omega_t}\widehat{f}_\theta - \widehat{u}_{\omega_t}b - \frac{1}{2}\widehat{u}_{\omega_t}^2 + \frac{\alpha}{2}\widehat{f}_\theta^2]$, obtained by replacing $f$ and $u$ in $\phi_t(\cdot)$ with their linearized counterparts defined in (17). The most important property of the linearized surrogate $\widehat{\phi}_t(\theta)$ is that it is *convex* in $\theta$. To estimate the rate of $\epsilon_f$, we start with the decomposition of regret. For any $\theta \in S_B$, define the regret $\mathrm{Reg}(\theta) = \frac{1}{T}\sum_{t=1}^T \phi_t(\theta_t) - \frac{1}{T}\sum_{t=1}^T \phi_t(\theta)$. Then we have the decomposition

$$\mathrm{Reg}(\theta) = \underbrace{\frac{1}{T}\sum_{t=1}^T \phi_t(\theta_t) - \frac{1}{T}\sum_{t=1}^T \widehat{\phi}_t(\theta_t)}_{(19)} + \underbrace{\frac{1}{T}\sum_{t=1}^T \widehat{\phi}_t(\theta_t) - \frac{1}{T}\sum_{t=1}^T \widehat{\phi}_t(\theta)}_{(20)} + \underbrace{\frac{1}{T}\sum_{t=1}^T \widehat{\phi}_t(\theta) - \frac{1}{T}\sum_{t=1}^T \phi_t(\theta)}_{(21)}.$$

(22)

We bound each term separately. To control the terms (19) and (21) we use the linearization of NN, which shows that the linearized NN and the original one behave similarly in terms of output and gradient as the width of NN $m$ grows (cf. Lemma 5.1 and Lemma B.1). The term (20) is bounded using techniques in convex online learning. The idea is to treat the algorithm designed for solving min-max game associated with $\phi$ as a *biased* primal-dual gradient methods for the one with $\widehat{\phi}$. We illustrate our techniques in further details in Appendix C.

# 6   Conclusions

We have derived saddle-point formulation for a class of generalized SEMs and parametrized the players with NNs. We show that the gradient-based primal-dual update enjoys global convergence in the overparametrized regimes ($m \to \infty$), for both 2-layer NNs and multi-layer NNs. Our results shed new light on the theoretical understanding of structural estimation with neural networks.

## Broader Impact

In recent years, the impact of machine learning (ML) on economics is already well underway [5, 15], and our work serves as a complement to this line of research. On the one hand, machine learning methods such as random forest, support vector machines and neural networks provide great flexibility in modeling, while traditional tools in structural estimation that are well versed in the econometrics community are still primitive, despite recent advances [32, 26, 7, 21]. On the other hand, to facilitate ML-base decision making, one must be aware of the distinction between prediction and causal inference. Our method provides an NN-based solution to estimation of generalized SEMs, which encompass a wide range of econometric and causal inference models. However, we remark that in order to apply the method to policy and decision problems, one must pay equal attention to other aspects of the model, such as interpretability, robustness of the estimates, fairness and nondiscrimination, assumptions required for model identification, and the testability of those assumptions. Unthoughtful application of ML methods in an attempt to draw causal conclusions must be avoided for both ML researchers and economists.

## Acknowledgments and Disclosure of Funding

This work is partially supported by the William S. Fishman Faculty Research Fund at the University of Chicago Booth School of Business. This work was completed in part with resources supported by the University of Chicago Research Computing Center.

We thank Professor Xiaohong Chen at Yale for pointing us to some of the classic works in nonparametric approach to and the use of NN in conditional moment problems, that were omitted in the submission version of this paper.

## Footnotes

[1]See Appendix E for a discussion of when a conditional expectation operator is compact.

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
