[Supplementary Material]

# APPENDICES TO PROVABLY EFFICIENT NEURAL ESTIMATION OF STRUCTURAL EQUATION MODEL: AN ADVERSARIAL APPROACH

## A  Examples of generalized structural equation models

In Section 2, we introduce our model in its full generality. Here we specialize it in concrete examples from the causal inference literature and econometrics.

We remark that the convergence result detailed in Theorem 4.1 applies to all examples while consistency result (Theorem 4.2) applies only to Example 1 because compactness of the conditional expectation operator is required in Theorem 4.2.

We add that the paper by Babii and Florens [6, Page 5, Footnote 4] includes a battery economics models that involve conditional moment restrictions, including the measurement error models, dynamic models with unobserved state variables, demand models, neoclassical trade models, models of earnings and consumption dynamics, structural random coefficient models, discrete games, models of two-sided markets, high-dimensional mixed-frequency IV regressions, and functional regression models. We refer readers to the paper for detailed references.

**Example 1, revisited** (*Instrumental Variable Regression*, [38, 26, 28]). In applied econometrics, endogeneity in regressors usually arises from omitted variables, measurement error, and simultaneity [50]. The method of instrumental variables (IV) provides a general solution to the problem of an endogenous explanatory variable. Without loss of generality, consider the model of the form

$$Y = g_0(X) + \varepsilon, \quad \mathbb{E}[\varepsilon \mid Z] = 0, \qquad \text{(2 revisited)}$$

where $g_0$ is the unknown function of interest, $Y$ is an observable scalar random variable, $X$ is a vector of explanatory variables, $Z$ is a vector of instrument variables, and $\varepsilon$ is the noise term. For the special case $X = Z$, the estimation of $g_0$ reduces to simple nonparametric regression, since $\mathbb{E}[Y \mid X = x] = g_0(x)$, and can be solved via spline regression or kernel regression [49]. When $X$ is endogenous, which is usually the case in observational data, traditional prediction-based methods fail to estimate $g_0$ consistently. In this case, $g_0(x) \neq \mathbb{E}[Y \mid X = x]$, and prediction and counterfactual prediction become different problems.

To see how the model fits our framework, define the operator $A : L^2(X) \to L^2(Z)$, $Ag = \mathbb{E}[g(X) \mid Z]$. Let $b = \mathbb{E}[Y \mid Z] \in L^2(Z)$. The structural equation (2) can be written as $Ag = b$. The minimax problem with penalty level $\alpha$ ($\alpha > 0$) takes the form

$$\min_{f \in L^2(X)} \max_{u \in L^2(Z)} \mathbb{E}[f(X)u(Z) - Y \cdot u(Z) - \tfrac{1}{2}u^2(Z) + \tfrac{\alpha}{2}f^2(X)], \qquad \text{(23)}$$

where the expectation is taken over all random variables.

The IV framework enjoys a long history, especially in economics [23]. It provides a means to answer counterfactual questions like what is the efficacy of a given drug in a given population? What fraction of crimes could have been prevented by a given policy? However, the presence of confounders makes these questions difficult. If $X$ is endogenous, which is usually the case in observational data, then $g_0(x) \neq \mathbb{E}[Y \mid X = x]$, and prediction and counterfactual prediction become different problems. When valid IVs are identified, we have a hope to answer these counterfactual questions.

Counterfactual prediction targets the quantity $\mathbb{E}[Y \mid \mathrm{do}(X = x)]$ defined by the causal graph (see Figure 1), where the $\mathrm{do}(\cdot)$ operator indicates that we have intervened to set the value of variable $X$ to $x$ while keeping the distribution of $\varepsilon$ fixed [41]. To facilitate counterfactual prediction, we need to impose stronger conditions on the model [36, 26]: (i) relevance: $\mathbb{P}(X \mid Z = z)$ is not constant in $z$; (ii) exclusion: $Y \perp\!\!\!\perp Z \mid X, \varepsilon$; and (iii) unconfounded instrument: $\varepsilon \perp\!\!\!\perp Z$. Figure 1 encodes such assumptions succinctly.

**Example 2, revisited** (*Simultaneous Equations Models*). Dynamic models of agent's optimization problems or of interactions among agents often exhibit simultaneity. Demand and supply model is such an example. Let $Q$ and $P$ denote the quantity sold and price of a product. Consider the demand and supply model adapted from [33].

$$\begin{aligned}
Q &= D(P, I) + U_1, \\
P &= S(Q, W) + U_2, \\
\mathbb{E}[U_1 \mid I, W] &= 0, \ \mathbb{E}[U_2 \mid I, W] = 0.
\end{aligned} \qquad \text{(3 revisited)}$$

Figure 1: A causal diagram of IV. Three observable variables $X, Y, Z$ (denoted by filled circles) and one unobservable confounding variable $\varepsilon$. There is no direct effect of the instrument $Z$ on the outcome $Y$ except through $X$.

Here $D$ and $S$ are functions of interest, $I$ denotes consumers' income, $W$ denotes producers' input prices, $U_1$ denotes an unobservable demand shock, and $U_2$ denotes an unobservable supply shock. Equation (3) is generally the results of equilibrium. Due to simultaneity, there is no hope to recover demand function $D$ by simple nonparametric regression of $Q$ on $P$ and $I$; nor can we recover supply function $S$ by regressing $P$ on $Q$ and $W$. The knowledge of $D$ is essential in predicting the effect of financial policy. For example, let $\tau$ be a percentage tax paid by the purchaser. Then the resulting equilibrium quantity is the solution $\hat{Q}$ to the equation

$$\hat{Q} = D\big((1+\tau)(S(\hat{Q}, I) + U_1), W\big) + U_2.$$

To cast the model (3) to a minimax problem, define the operators

$$A_1 : L^2(P, I) \to L^2(I, W), A_1 D = \mathbb{E}[D(P, I) \mid I, W],$$
$$A_2 : L^2(Q, W) \to L^2(I, W), A_2 S = \mathbb{E}[S(Q, W) \mid I, W].$$

The resulting minimax problem is

$$\min_{\substack{D \in L^2(P, I), \\ S \in L^2(Q, W)}} \max_{u_1, u_2 \in L^2(I, W)} \left\{ \begin{matrix} \mathbb{E}[u_1(I,W) \cdot (D(P,I) - Q) + u_2(I,W) \cdot (S(Q,W) - P)] \\ -\frac{1}{2} u_1(I,W)^2 - \frac{1}{2} u_2(I,W)^2] \end{matrix} \right\}.$$

Note in this case the operators $A_1$ and $A_2$ are not compact [11] due to common elements. The min-max derivation remains valid but the stability of the solution is left for future work.

The causal reading of the simultaneous equations models is an open question since an important assumption often made in causal discovery is that the causal mechanism is acyclic, i.e., that no feedback loops are present in the system [41]. There are efforts in bridging this gap; see, for example, [35].

**Example 3, revisited** (*Dynamic Panel Data Model*, [46]). Panel data is a common form of econometric data; it contains observations of multiple units measured over multiple time periods. We consider the dynamic model of the following form that includes time-varying regressors, allowing us to investigate the long-run relationship between economic factors [46].

$$Y_{it} = m\left(Y_{i,t-1}, X_{it}\right) + \alpha_i + \varepsilon_{it}, \qquad \text{(4 revisited)}$$
$$\mathbb{E}[\varepsilon_{it} \mid \underline{Y}_{i,t-1}, \underline{X}_{it}] = 0, \quad i = 1, \dots, N, \quad t = 1, \dots, T,$$

where $X_{it}$ is a $p \times 1$ vector of regressors, $m$ is the unknown function of interest, $\alpha_i$'s are the unobserved individual-specific fixed effects, potentially correlated with $X_{it}$, and $\varepsilon_{it}$'s are idiosyncratic errors. $\underline{X}_{it} := (X_{it}^\top, \dots, X_{i1}^\top)^\top$ and $\underline{Y}_{i,t-1} := (Y_{i,t-1}, \dots, Y_{i1})^\top$ are the history of individual $i$ up to time $t$. We assume that $(Y_{it}, X_{it}, \varepsilon_{it})$ are i.i.d. along the individual dimension $i$ but may not be strictly stationary along the time dimension $t$. Clearly, for a large $t$ the conditional set $\{\underline{Y}_{i,t-1}, \underline{X}_{it}\}$ contains a large number of valid instruments. We do not pursue a search for an efficient choice of IVs in the paper.

To see how it relates to model (1), we consider the first-differenced model

$$\Delta Y_{it} = m\left(U_{i,t-1}\right) - m\left(U_{i,t-2}\right) + \Delta\varepsilon_{it}, \qquad (24)$$
$$\mathbb{E}[\Delta\varepsilon_{it} \mid U_{i,t-2}] = 0, \quad i = 1, \dots, N, \quad t = 3, \dots, T, \qquad (25)$$

where $\Delta Y_{it} := Y_{it} - Y_{i,t-1}$, $U_{i,t-2} := [Y_{i,t-2}, X_{i,t-1}^\top]^\top$ and $\Delta\varepsilon_{it} := \varepsilon_{it} - \varepsilon_{i,t-1}$. The conditional expectation (25) is obtained by applying law of iterated expectation to (4) conditional on $U_{i,t-2}$.

Model (24) cannot be solved via traditional nonparametric regression because $\Delta\varepsilon_{it}$ is generally correlated with $Y_{i,t-1}$ on the RHS of (24).

Now we cast the model (25) into a minimax problem. For ease of exposition we assume strict stationarity on the sequence $\{U_{it}\}$, which implies that the marginal distribution of $U_{i,t-1}$ and the transition distribution $p(U_{i,t-1}|U_{i,t-2})$ are time-invariant. Now we define a random vector $(D', E', D, E, F, \varepsilon) =_d (Y_{i,t-1}, X_{it}, Y_{i,t-2}, X_{i,t-1}, \Delta Y_{it}, \Delta\varepsilon_{it})$, and the definition is valid due to stationarity. Equation (25) can be rewritten as

$$\mathbb{E}[F - m(D', E') + m(D, E) \mid E, D] = 0.$$

Define the operator $A : L^2(D', E') \to L^2(E, D)$, $Am = \mathbb{E}[m(D', E') \mid E, D]$ and the function $b = \mathbb{E}[F \mid E, D]$. Equation (25) becomes $(A - I)m = b$, which is a Frehdolm equation of type II. The key difference between type I and type II Fredholm equations lies in stability of the solution. If $I - K : \mathcal{H} \to \mathcal{H}$ is injective, then it is surjective, the inverse operator $(I - K)^{-1}$ is continuous and therefore the solution to type II equation is stable [30].

We remark that $1$ is the greatest eigenvalue of $A$ because $(D', E')$ and $(D, E)$ are identically distributed. Therefore we assume the multiplicity of $1$ is one in order to identify $m$ up to a constant. The resulting min-max problem is

$$\min_{m \in L^2(D', E')} \max_{u \in L^2(E, D)} \mathbb{E}[u(E, D) \cdot \big(F - m(D', E') + m(D, E)\big) - \tfrac{1}{2}u(E, D)^2].$$

In the absence of the lagged term $Y_{i,t-1}$ on the RHS of (4), the model (4) reduces to the nonparametric panel data model [27],

$$Y_{it} = m\left(X_{it}\right) + \alpha_i + \varepsilon_{it}, \quad i = 1, \ldots, n, \quad t = 1, \ldots, T.$$

If the lag term does not appear, we recover the measurement error model studied in [11].

**Example 4** (Euler Equation and Utility, [20])**.** In economic models, the behavior of an optimizing agent can be characterized by Euler equations [25]. Consumption-based capital asset pricing model (CCAPM) is such an example. Here we consider a simplified setting of [20] where at time $t$ an agent receives income $W_t$ and purchases or sells certain units of an asset at price $P_t$. For simplicity we assume there is only one asset on the market. Let $U$ be a time-invariant utility function, and $b \in (0, 1)$ be the discount factor. $U$ and $b$ are parameters of interests known to the agent but unknown to the researchers. The stream of consumption $\{C_t\}$ is the solution to the optimization problem

$$\max_{\{C_t, Q_t\}_{t=0}^\infty} \mathbb{E}\left[\sum_{t=0}^\infty \beta^t U(C_t)\right] \tag{26}$$

$$\text{s.t. } C_t + P_t Q_t = P_t Q_{t-1} + W_t, \tag{27}$$

where $Q_t$ is the quantity of the asset owned by the agent at time $t$. RHS of the constraint (27) is the total value owned by the agent before the exchange at time $t$, while the LHS represents the total value after the exchange. The agent manipulates his consumption, $C_t$, and the quantity of the asset he holds, $Q_t$, to maximize his expected long-run discounted utility.

Define $R_t = P_{t+1}/P_t$. Using the method of Lagrange multiplier, one can obtain the optimality condition of (26)

$$\mathbb{E}\left[R_{t+1}\beta\frac{U'(C_{t+1})}{U'(C_t)} - 1 \mid I_t\right] = 0, \tag{28}$$

where $I_t$ represents the information available at time $t$. A derivation can be found in [20]. Let $g = U'$ be the marginal utility function. Conditioning on $C_t$ in (28), we obtain

$$\mathbb{E}[\beta R_{t+1} g(C_{t+1}) \mid C_t] = g(C_t). \tag{29}$$

The goal to estimate the function $g$ given $\{C_t, R_{t+1}, C_{t+1}\}$. To see how our min-max derivation applies, define the operator $A : L^2(C_{t+1}) \to L^2(C_t)$, $(Ag)(c) = \mathbb{E}[g(C_{t+1})R_{t+1} \mid C_t = c]$. We assume $A$ is well-defined. Then (28) can be succinctly written as

$$\beta Ag = g.$$

We remark that $g$ is identified up to an arbitrary sign and scale normalization; [20] provides a detailed discussion on identification. Assuming $\beta$ is known, the resulting min-max problem is

$$\min_{g \in L^2(C_{t+1})} \max_{u \in L^2(C_t)} \mathbb{E}\big[\beta g(C_{t+1})R_{t+1}u(C_t) - g(C_t)u(C_t) - \tfrac{1}{2}u^2(C_t)\big].$$

One caveat is that $g = 0$ is a trivial solution to (29) and therefore during the training of NNs we should avoid such a solution. The empirical performance of Algorithm 1 in this example is left for future work.

**Example 5** (Proxy Variables of an Unmeasured Confounder, [34]). Consider the causal DAG in Figure 2 in the sense of Pearl [41]. Here $X$ and $Y$ denote the treatment and the outcome, respectively. The confounder $U$ is unobserved, while its proxies $Z$ and $W$ are observed. Assume $U, W, Z$ are continuous and in the discussion we assume $X$ and $Y$ are fixed at $(x, y)$. The conditional independence encoded in Figure 2 is $W \perp\!\!\!\perp (Z, X) \mid U$ and $Z \perp\!\!\!\perp Y \mid (U, X)$. Using the do-operator of Pearl [41], the causal effect of $X$ on $Y$ is

$$p(y \mid \mathrm{do}(x)) = \int p(y|x, u)p(u)du,$$

where $p(\cdot)$ stands for probability mass functions of a discrete variable or the probability density function for a continuous variable. However, $U$ is unobserved so we cannot directly calculate the causal effect.

The work of Miao et al. [34] provides an identification strategy for the causal effect of $X$ on $Y$ with the help of the confounder proxies $Z$ and $W$. Consider the solution $h(w, x, y)$ to the following integral solution: for all $(x, y)$ and for all $z$,

$$p(y|z, x) = \int_{-\infty}^{+\infty} h(w, x, y)p(w|z, x)dw, \tag{30}$$

which is a Fredholm integral equation of the first kind.

**Lemma A.1** (Theorem 1 of [34]). *Assume the causal DAG in Figure 2 and that a solution to* (30) *exists. Assume the following completeness condition:* $\mathbb{E}[g(U)|Z, X] = 0$ *almost surely if and only if* $g(u) = 0$ *almost surely. Then* $p(y \mid \mathrm{do}(x)) = \int_{-\infty}^{+\infty} h(w, x, y)p(w)dw$.

The result suggests that one can identify the causal effect by first solving for $h$ in (30) and then applying Lemma A.1, since $p(y|z, x)$, $p(w|z, x)$ and $p(w)$ can be estimated from the data. To see how (30) fits into our framework, we note that Equation (30) implies $\mathbb{E}[\mathbb{1}\{Y = y\} \mid Z, X] = \mathbb{E}[h(W, X, y) \mid Z, X]$ for all $y$, and thus similar min-max problem derivation goes through. However, in [34] the identification strategy is limited to the case where $X$ and $Y$ are categorical, and it would be interesting to see how our method performs in the setting of continuous treatment and continuous outcome.

Figure 2: A causal graph of confounder proxies. Adapted from Figure 1(f) of [34].

# B   Linear approximation error of multi-layer NNs

Without assumptions on the distribution of data (Assumption A.4), we have slightly worse upper bounds on the error of linearization for multi-layer NNs.

**Lemma B.1** (Error of local linearization, multi-layer, [2, 22]). *Consider the multi-layer neural networks described in* (10). *Under Assumption A.2, with probability at least* $1 - \exp(-\Omega(\log^2 m))$ *with respect to the random initialization, for any* $W \in S_{B,H}$ *and all* $x$ *such that* $\|x\| = 1$,

1. $|\widehat{f}(x, W)| = \mathcal{O}(BH^{3/2} \log m)$,

2. $\|\nabla_W f(x, W)\| = \mathcal{O}(H)$,

3. $|f(x, W) - \widehat{f}(x, W)| = \mathcal{O}(B^{4/3} m^{-1/6} H^3 \log^{1/2} m)$, *and*

4. $\|\nabla_W f(x, W) - \nabla_W \widehat{f}(x, W)\| = \mathcal{O}(B^{1/3} m^{-1/6} H^{5/2} \log^{1/2} m)$.

*Proof.* See Section D.2 in Appendix D. □

## C  Bounds on the terms (19), (20) and (21)

### C.1  Bounds on the terms (19), (21)

First, we establish the closeness between the original function $\phi$ and the one consists of linearized NNs, $\widehat{\phi}$. The following lemma shows that $\widehat{\phi}$ is a good surrogate for $\phi$ in the sense that the approximation error is of order $\mathcal{O}(aB^{5/2}m^{-1/4})$, which vanishes as $m \to \infty$.

Denote $F(\theta, \omega; X_1, X_2) = u_\omega f_\theta - u_\omega b - \frac{1}{2} u_\omega^2 + \frac{\alpha}{2} f_\theta^2$. Note $\mathbb{E}_X [F(\theta, \omega; X_1, X_2)] = \phi(\theta, \omega)$. Similarly we define $\widehat{F}(\theta, \omega; X_1, X_2) = \widehat{u}_\omega \widehat{f}_\theta - \widehat{u}_\omega b - \frac{1}{2} \widehat{u}_\omega^2 + \frac{\alpha}{2} \widehat{f}_\theta^2$.

**Lemma C.1** (Closeness between $\widehat{\phi}$ and $\phi$). *Let $a = \max\{1, \alpha\}$. For any $\theta, \omega \in S_B$, we have*

$$\mathbb{E}_{\text{init}} \big[ |\widehat{\phi}(\theta, \omega) - \phi(\theta, \omega)| \big] = \mathcal{O}\big(aB^{5/2} m^{-1/4}\big).$$

*Proof.* See Section D.4 in Appendix D. The proof relies on the decay rates of approximation error, as detailed in Lemma 5.1. □

Lemma C.1 suggests it suffices to set

$$\epsilon_f = \mathcal{O}(aB^{5/2} m^{-1/4}) + \max_\theta \Big( \frac{1}{T} \sum_{t=1}^T \widehat{\phi}_t(\theta_t) - \frac{1}{T} \sum_{t=1}^T \widehat{\phi}_t(\theta) \Big). \tag{31}$$

We now turn to bound the term (20) using techniques adapted from convex online learning analysis.

### C.2  A bound on the term (20)

We emphasize we apply online learning analysis (Lemma C.2) to the regret associated with $\widehat{\phi}_t$'s but using updates designed for $\phi_t$'s.

**Lemma C.2** (Online convex learning with noisy and biased gradient). *Given a sequence of convex functions on a convex space $\Theta$, $f_1, f_2, \cdots : \Theta \to \mathbb{R}$, consider the projected gradient descent updates*

$$\theta_{t+1} = \Pi_\Theta \big( \theta_t - \eta \left( \zeta_t + \xi_t \right) \big), \tag{32}$$

*where $\mathbb{E}\left[\zeta_t | \theta_t\right] = \nabla f_t(\theta_t)$, $\Pi_\Theta(\theta) \in \arg\max_{\theta' \in \Theta} \|\theta - \theta'\|$ is the projection map to $\Theta$. Assume $\sup_t \|\zeta_t + \xi_t\| < K$ a.s. and $\sup_\theta \|\theta\| < M$. Then with probability at least $1 - \delta$,*

$$\frac{1}{T} \sum_{t=1}^T f_t(\theta_t) - \frac{1}{T} \sum_{t=1}^T f_t(\theta) \leqslant \frac{\eta K}{2} + \frac{M}{T\eta} + 8K \sqrt{\frac{M \ln(1/\delta)}{T}} + \frac{2\sqrt{2M}}{T} \sum_{t=1}^T \|\xi_t\| \tag{33}$$

*for all $\theta \in \Theta$.*

*Proof.* See Section D.5 in Appendix D. □

In order to apply Lemma C.2 to analyze the regret generated by the sequence $\{\widehat{\phi}_t\}$ with actual updates being $\nabla_\theta F_t(\theta_t; X_{1,t}, X_{2,t})$ instead of $\nabla_\theta \widehat{\phi}_t(\theta_t)$, we need to verify two conditions: (i) bounded update steps, i.e., $\|\nabla_\theta F_t(\theta_t; X_{1,t}, X_{2,t})\|$ is bounded for all $t$, and (ii) bounded parameter space.

To achieve global convergence, we also require that bias in updates, $\|\nabla_\theta F_t(\theta_t; X_{1,t}, X_{2,t}) - \nabla_\theta \widehat{\phi}_t(\theta_t)\|$, which corresponds to the $\|\xi_t\|$ term in (33), converges to zero as $m \to \infty$. In our analysis we assume $\nabla_\theta \widehat{F}_t(\theta; X_{1,t}, X_{2,t})$ is an unbiased estimate of $\nabla_\theta \widehat{\phi}_t(\theta)$. The following lemma summarizes the results we need to apply Lemma C.2 and obtain a bound on the term (20).

**Lemma C.3** (Bounded gradient and vanishing bias). *Consider the updates in algorithm (Algorithm 1). For all $\omega_t$, $\theta$, the following holds.*

1. $\|\nabla_\theta F_t(\theta; x_1, x_2)\| = \mathcal{O}(aB)$ *for all* $x, y$, *and*

2. $\mathbb{E}_{\text{init}, X}\big[\|\nabla_\theta F(\theta, \omega_t; X_1, X_2) - \nabla_\theta \widehat{F}(\theta, \omega_t; X_1, X_2)\|\big] = \mathcal{O}(aB^{3/2}m^{-1/4})$.

*Proof.* See Section D.6 in Appendix D. $\qquad\qquad\qquad\qquad\qquad\qquad\qquad\qquad\qquad$ □

Equipped with Lemma C.1 and Lemma C.3, we are now ready to obtain a bound on the regret $\epsilon_f$ defined in (18). Set $M = B$, $K = aB$, $\|\xi_t\| = \mathcal{O}(aB^{3/2}m^{-1/4})$ in the RHS of (33), continue (31), and we obtain with probability at least $1 - \delta$ with respect to sampling process,

$$\mathbb{E}_{\text{init}}[\epsilon_f] = \underbrace{\mathcal{O}\big(aB^{5/2}m^{-1/4}\big)}_{\text{linearization error (19) and (21)}} + \underbrace{\mathcal{O}\Big(\frac{a\eta B}{2} + \frac{B}{T\eta} + \frac{aB^{3/2}\log^{1/2}(1/\delta)}{T^{1/2}} + \frac{aB^4}{m^{1/4}}\Big)}_{\text{optimization error (20)}}.$$

It can be shown $\epsilon_u$ is of the same order, thus completing the proof of claim 1 in Theorem 4.1.

# D    Proof of theorems

**A remark on notations.** Throughout the proof we ignore dependence on $\theta, \omega, X_1, X_2$ and the NN initial parameters $\Xi_0$ or $\Xi_{H,0}$ defined in (9) and (10), respectively. For readers' convenience, we now restate the dependence of all the functions on their parameters. Recall the NN $f_\theta(X_1) = f(\theta; X_1)$ is an NN with weights $\theta$ and input $X_1$ and similarly for $u_\omega(X_2) = u(\omega; X_2)$. Note $f_\theta$ and $u_\theta$ depend on the initialization implicitly through the range of NN weights (which is centered around the initial weight) and the output layer weights (and the input layer weight, too, in the case of multi-layer NNs). Recall

$$\phi = \phi(\theta, \omega) = \phi(f_\theta, u_\omega) := \mathbb{E}\big[\big(f(\theta; X_1) - b(X_2)\big)u(\omega; X_2) + \tfrac{\alpha}{2}f(\theta; X_1)^2 - \tfrac{1}{2}u(\omega; X_2)^2\big],$$

and

$$F = F(\theta, \omega; X_1, X_2) = \big(f(\theta; X_1) - b(X_2)\big)u(\omega; X_2) + \tfrac{\alpha}{2}f(\theta; X_1)^2 - \tfrac{1}{2}u(\omega; X_2)^2,$$

and they satisfy $\phi(\theta, \omega) = \mathbb{E}_{X_1, X_2}[F(\theta, \omega; X_1, X_2)]$. Note $\phi$ is convex-concave in $(f, u)$ but not in $(\theta, \omega)$. Recall the linearized counterparts of $f$ and $u$, defined in (17), are $\widehat{f}_\theta = \widehat{f}(\theta(0); X_1) + \langle \nabla_\theta f(\theta(0), X_1), \theta - \theta(0)\rangle$ and similarly for $\widehat{u}_\omega$. Now we replace NNs $f_\theta$ and $u_\omega$ by their hat-versions in the definition of $\phi$ and $F$ and obtain $\widehat{\phi} = \widehat{\phi}(\theta, \omega, \Xi_0)$, and $\widehat{F} = \widehat{F}(\theta, \omega, \Xi_0; X_1, X_2)$. In the proof we only discuss the case where $b = b(X_2)$ is known. The proof goes thorough for the more general case $b(X_2) = \mathbb{E}[\widetilde{b}(X_1, X_2) \mid X_2]$ with little modifications.

## D.1    Proof of Lemma 5.1

*Proof.* The proof follows closely Lemma 5.1 and Lemma 5.2 in [10]. Recall that the weights of a 2-layer NN is represented by $W \in \mathbb{R}^{md}$ where $d$ is the input dimension and $m$ is the number of neurons. $W_r \in \mathbb{R}^d$ represents the weights connecting inputs and the $r$-th neuron. $W = [W_1^\top, \dots, W_r^\top]^\top$.

We start with

$$\|\nabla_W f(x; W)\|_2^2 \leqslant \frac{1}{m}\sum_{r=1}^m \mathbb{1}\left\{W_r^\top x > 0\right\}\|x\|_2^2 \leqslant 1$$

for all $W \in S_B$, all $x$. So claim 2 follows. Claim 1 is indeed true because $f(x, W)$ is 1-Lipschitz wrt $W$ and that $\|W - W(0)\|_2 \leqslant B$ for all $W \in S_B$. To show claim 3 we first analyze the expression

$|f(x, W) - \widehat{f}(x, W)|.$

$$|f(x, W) - \widehat{f}(x, W)|$$

$$= \frac{1}{\sqrt{m}} \left| \sum_{r=1}^{m} \left( \mathbb{1}\left\{W_r^\top x > 0\right\} - \mathbb{1}\left\{W_r(0)^\top x > 0\right\} \right) \cdot b_r W_r^\top x \right|$$

$$\leqslant \frac{1}{\sqrt{m}} \sum_{r=1}^{m} \left| \mathbb{1}\left\{W_r^\top x > 0\right\} - \mathbb{1}\left\{W_r(0)^\top x > 0\right\} \right| \cdot \left( \left|W_r(0)^\top x\right| + \|W_r - W_r(0)\|_2 \right)$$

$$\leqslant \frac{1}{\sqrt{m}} \sum_{r=1}^{m} \mathbb{1}\left\{ \left|W_r(0)^\top x\right| \leqslant \|W_r - W_r(0)\|_2 \right\} \cdot \left( \left|W_r(0)^\top x\right| + \|W_r - W_r(0)\|_2 \right)$$

$$\leqslant \frac{2}{\sqrt{m}} \sum_{r=1}^{m} \mathbb{1}\left\{ \left|W_r(0)^\top x\right| \leqslant \|W_r - W_r(0)\|_2 \right\} \cdot \|W_r - W_r(0)\|_2. \tag{34}$$

Here the first inequality follows from $\|x\|_2 = 1$. The second inequality follows from the following reasoning.

$$\mathbb{1}\left\{W_r^\top x > 0\right\} \neq \mathbb{1}\left\{W_r(0)^\top x > 0\right\}$$
$$\implies \left|W_r(0)^\top x\right| \leqslant \left|W_r^\top x - W_r(0)^\top x\right| \leqslant \|W_r - W_r(0)\|_2.$$

The third inequality follows from $\mathbb{1}\{|x| \leqslant y\}|x| \leqslant \mathbb{1}\{|x| \leqslant y\}y$ for all $x, y > 0$.

Next we square both sides of (34), invoke Cauchy-Schwartz inequality, and the fact that $\|W - W(0)\|_2 \leqslant B$.

$$|f(x, W) - \widehat{f}(x, W)|^2 \leqslant \frac{4B^2}{m} \sum_{r=1}^{m} \mathbb{1}\left\{ \left|W_r(0)^\top x\right| \leqslant \|W_r - W_r(0)\|_2 \right\}. \tag{35}$$

To control the expectation of the RHS of (35), we introduce the following lemma.

**Lemma D.1.** *There exists a constant $c_1 > 0$, such that for any random vector $W$ such that $\|W - W(0)\|_2 \leqslant B$, it holds that*

$$\mathbb{E}_{\text{init},x}\left[ \frac{1}{m} \sum_{r=1}^{m} \mathbb{1}\left\{ \left|W_r(0)^\top x\right| \leqslant \|W_r - W_r(0)\|_2 \right\} \right] \leqslant c_1 B \cdot m^{-1/2}.$$

Taking expectation on both sides of (35) we get

$$\mathbb{E}_{\text{init},x}\left[ |f(x, W) - \widehat{f}(x, W)|^2 \right] \leqslant 4c_1 B^3 \cdot m^{-1/2},$$

establishing claim 3. Claim 4 also follows from Lemma D.1 as follows.

$$\|\nabla_W f(x, W) - \nabla_W \widehat{f}(x, W)\|_2^2$$

$$= \frac{1}{m} \sum_{r=1}^{m} \left( \mathbb{1}\left\{W_r^\top x > 0\right\} - \mathbb{1}\left\{W_r(0)^\top x > 0\right\} \right)^2 \cdot \|x\|_2^2$$

$$\leqslant \frac{1}{m} \sum_{r=1}^{m} \mathbb{1}\left\{ \left|W_r(0)^\top x\right| \leqslant \|W_r - W_r(0)\|_2 \right\}.$$

$\square$

**Proof of Lemma D.1**

*Proof.* The proof follows Lemma H.1 of [10] and is stated for completeness. By the assumption that there exists $c_0 > 0$, for any unit vector $v \in \mathbb{R}^d$ and any constant $\zeta > 0$, such that $\mathbb{P}_X\left(|v^\top X| \leqslant \zeta\right) \leqslant c\zeta$, we have

$$\mathbb{E}_{\text{init},x}\left[ \frac{1}{m} \sum_{r=1}^{m} \mathbb{1}\left\{ \left|W_r(0)^\top x\right| \leqslant \|W_r - W_r(0)\|_2 \right\} \right]$$

$$\leqslant \mathbb{E}_{\text{init}}\left[ \frac{1}{m} \sum_{r=1}^{m} c_0 \cdot \|W_r - W_r(0)\|_2 / \|W_r(0)\|_2 \right]. \tag{36}$$

Note the expectation in (36) does not involve the data distribution. Next we apply Hölder's inequality.

$$
\mathbb{E}_{\text{init},x}\left[\frac{1}{m}\sum_{r=1}^{m}\mathbb{1}\left\{\left|W_r(0)^\top x\right| \leqslant \|W_r - W_r(0)\|_2\right\}\right]
$$

$$
\leqslant c_0/m \cdot \mathbb{E}_{\text{init}}\left[\left(\sum_{r=1}^{m}\|W_r - W_r(0)\|_2^2\right)^{1/2} \cdot \left(\sum_{r=1}^{m}\frac{1}{\|W_r(0)\|_2^2}\right)^{1/2}\right]
$$

$$
\leqslant c_0 B m^{-1} \cdot \mathbb{E}_{\text{init}}\left[\sum_{r=1}^{m}\frac{1}{\|W_r(0)\|_2^2}\right]^{1/2}
$$

$$
\leqslant c_0 B m^{-1} \cdot \sqrt{m} \cdot \mathbb{E}_{w\sim N(0,I_d/d)}\left[1/\|w\|_2^2\right]^{1/2}.
$$

Setting $c_1 = c_0 \cdot \mathbb{E}_{w\sim N(0,I_d/d)}\left[1/\|w\|_2^2\right]^{1/2}$ finishes the proof. $\qquad\square$

### D.2 Proof of Lemma B.1

*Proof.* See [3, 22] for a detailed proof. Also see Appendix F in [10]. In detail, claim 1 follows from equation F.10 of [10]. Claim 2 and claim 4 follow from Lemma F.1 of [10]. Claim 3 follows from Lemma F.2 of [10]. $\qquad\square$

### D.3 Proof of Lemma 5.2

*Proof.* Recall $\phi(f,u)$ is convex in $f$ and concave in $u$, and that $L(f)$ is convex in $f$. The final output $\bar{f}_T$ is the average of the sequence $\{f_t\}_{t=1}^{T}$ and so is $\bar{u}_T$. Recall $\epsilon_f, \epsilon_u$ satisfy

$$
\frac{1}{T}\sum_{t=1}^{T}\phi(f_t, u_t) \leqslant \min_{f\in\mathcal{F}_{NN}}\frac{1}{T}\sum_{t=1}^{T}\phi(f, u_t) + \epsilon_f,
$$

$$
\frac{1}{T}\sum_{t=1}^{T}\phi(f_t, u_t) \geqslant \max_{u\in\mathcal{F}_{NN}}\frac{1}{T}\sum_{t=1}^{T}\phi(f_t, u) - \epsilon_u.
$$

Note both $f$ and $u$ range over the space of NNs. We start with the equivalent expression for $L$ defined in (7). By Assumption A.5, for all $f \in \mathcal{F}_{NN}$, $L(f) = \max_{u\in\mathcal{F}_{NN}}\phi(f,u)$ with $\phi$ defined in (8). We have

$$
L(\bar{f}_T) - L^*
$$

$$
= \max_{u\in\mathcal{F}_{NN}}\phi(\bar{f}_T, u) - \min_{f\in\mathcal{F}_{NN}}\max_{u\in\mathcal{F}_{NN}}\phi(f, u)
$$

$$
\leqslant \max_{u\in\mathcal{F}_{NN}}\phi(\bar{f}_T, u) - \min_{f\in\mathcal{F}_{NN}}\phi(f, \bar{u}_T)
$$

$$
\leqslant \max_{u\in\mathcal{F}_{NN}}\frac{1}{T}\sum_{t=1}^{T}\phi(f_t, u) - \min_{f\in\mathcal{F}_{NN}}\frac{1}{T}\sum_{t=1}^{T}\phi(f, u_t)
$$

$$
= \left[\left(\max_{u\in\mathcal{F}_{NN}}\frac{1}{T}\sum_{t=1}^{T}\phi(f_t, u)\right) - \frac{1}{T}\sum_{t=1}^{T}\phi(f_t, u_t)\right]
$$

$$
+ \left[\left(\frac{1}{T}\sum_{t=1}^{T}\phi(f_t, u_t)\right) - \min_{f\in\mathcal{F}_{NN}}\frac{1}{T}\sum_{t=1}^{T}\phi(f, u_t)\right]
$$

$$
\leqslant \epsilon_f + \epsilon_u.
$$

In fact, we easily have $\frac{1}{T}\sum_{t=1}^{T}L(f_i) - L^* \leqslant \epsilon_f + \epsilon_u$. $\qquad\square$

### D.4 Proof of Lemma C.1

*Proof.* Recall $X = [X_1^\top, X_2^\top]^\top$, $\phi(\theta, \omega) = \mathbb{E}_X[F(\theta, \omega; X_1, X_2)] = \mathbb{E}_{XY}[uf - ub - (1/2)u^2 + (\alpha/2)f^2]$.

Denote $\widehat{F}(\theta, \omega) = \widehat{u}\widehat{f} - \widehat{u}b - (1/2)\widehat{u}^2 + (\alpha/2)\widehat{f}^2$, where the hat-version are the linearized NN. We start by noting

$$\mathbb{E}_{\text{init}}\big[|\widehat{\phi}(\theta, \omega) - \phi(\theta, \omega)|\big]$$
$$= \mathbb{E}_{\text{init}, X}\big[|\widehat{F} - F|\big]$$
$$= \mathbb{E}_{\text{init}, X}\big[|(\widehat{u}\widehat{f} - \widehat{u}b - \tfrac{1}{2}\widehat{u}^2 + \tfrac{\alpha}{2}\widehat{f}^2) - (uf - ub - \tfrac{1}{2}u^2 + \tfrac{\alpha}{2}f^2))|\big]$$
$$\leqslant \mathbb{E}_{\text{init}, XY}\big[|\widehat{u}\widehat{f} - uf|\big] + \mathbb{E}_{\text{init}, X}\big[|(\widehat{u} - u)b|\big] + (1/2)\mathbb{E}_{\text{init}, X}\big[|\widehat{u}^2 - u^2|\big] + (\alpha/2)\mathbb{E}_{\text{init}, X}\big[|\widehat{f}^2 - f^2|\big].$$

Now bound the terms

$$\mathbb{E}_{\text{init}, X}\big[|\widehat{u}\widehat{f} - uf|\big], \tag{37}$$
$$\mathbb{E}_{\text{init}, X}\big[|(\widehat{u} - u)b|\big], \tag{38}$$
$$\mathbb{E}_{\text{init}, X}\big[|\widehat{u}^2 - u^2|\big], \tag{39}$$
$$\mathbb{E}_{\text{init}, X}\big[|\widehat{f}^2 - f^2|\big]. \tag{40}$$

For the term (37), we have

$$\mathbb{E}_{\text{init}, X}\big[|\widehat{u}\widehat{f} - uf|\big]$$
$$\leqslant \mathbb{E}_{\text{init}, X}\big[|\widehat{u}(\widehat{f} - f)|\big] + \mathbb{E}_{\text{init}, X}\big[(\widehat{u} - u)f\big]$$
$$\leqslant \sqrt{\mathbb{E}_{\text{init}, X}\big[\widehat{u}^2\big]\mathbb{E}_{\text{init}, X}\big[|\widehat{f} - f|^2\big]} + \sqrt{\mathbb{E}_{\text{init}, X}\big[f^2\big]\mathbb{E}_{\text{init}, X}\big[|\widehat{u} - u|^2\big]}$$
$$\text{(Cauchy-Schwarz inequality)}$$
$$= \sqrt{\mathcal{O}(B^2 \cdot B^3 m^{-1/2})} + \sqrt{\mathcal{O}(B^3 m^{-1/2}) \cdot \mathcal{O}(B^2)} \qquad \text{(Lemma 5.1)}$$
$$= \mathcal{O}(B^{5/2}m^{-1/4}).$$

We can apply similar techniques and obtain the following bounds on (38) and (40).

$$\mathbb{E}_{\text{init}, X}\big[|(\widehat{u} - u)b|\big] = \mathcal{O}(B^{3/2}m^{-1/2}),$$
$$\mathbb{E}_{\text{init}, X}\big[|\widehat{u}^2 - u^2|\big] = \mathcal{O}(B^{5/2}m^{-1/4}).$$

Putting all pieces together we get

$$\mathbb{E}_{\text{init}}\big[|\widehat{\phi}(\theta, \omega) - \phi(\theta, \omega)|\big] = \mathcal{O}((1 + \alpha)B^{5/2}m^{-1/4}).$$

$\square$

## D.5 Proof of Lemma C.2

*Proof.* We need the following lemma that controls regret in the context of online learning with exact gradient, and then we extend it to our noisy and biased gradient scenario.

**Lemma D.2** (Regret analysis in online learning, [45]). *Let $f_1, f_1, \cdots : \Theta \to \mathbb{R}$ be convex functions, where $\Theta$ is convex. Consider the mirror descent updates,*

$$\zeta_{t+1} = \nabla h^* \left(\nabla h\left(\theta_t\right) - \eta\nabla f_t\left(\theta_t\right)\right),$$
$$\theta_{t+1} = \arg\min_{\theta \in \Theta} D_h\left(\theta, \zeta_{t+1}\right),$$

*where $h$ is 1-strongly convex with respect to the norm $\|\cdot\|$, $D_h(x, y) = h(x) - h(y) - \nabla h(y)^\top(x - y)$ is the Bregman divergence, $h^*$ is the convex conjugate of $h$, and $\|\cdot\|_*$ is the dual norm of $\|\cdot\|$. Suppose that $\sup_t \|\nabla f_t\left(\theta_t\right)\|_* < K$ and $\sup_\theta h(\theta) < M$. Then for all $\theta \in \Theta$,*

$$\frac{1}{T}\sum_{t=1}^{T} f_t\left(\theta_t\right) - \frac{1}{T}\sum_{t=1}^{T} f_t(\theta) \leqslant \frac{\eta K}{2} + \frac{M}{T\eta}.$$

We refer readers to [45] for a proof of Lemma D.2. Now we take $h(x) = \frac{1}{2}\|x\|$, and $\|x\|$ is the Euclidean norm.

Note that in our case the actual update is $\zeta_t + \xi_t$, where $\zeta_t$ is an unbiased estimate of the gradient $\nabla f_t(\theta_t)$, and $\xi_t$ is a noise term. We construct linear surrogate functions $\widehat{f}_t(\theta) = f_t(\theta_t) + (\zeta_t + \xi_t)^\top (\theta - \theta_t)$ and notice that $\zeta_t + \xi_t$ is indeed the gradient of the surrogate at $\theta_t$, i.e., $\nabla \widehat{f}_t(\theta_t) = \zeta_t + \xi_t$. Now we apply Lemma $D.2$ to the sequence $\{\widehat{f}_t\}$ and obtain

$$\frac{1}{T} \sum_{t=1}^{T} \widehat{f}_t(\theta_t) - \frac{1}{T} \sum_{t=1}^{T} \widehat{f}_t(\theta) \leqslant \frac{\eta B}{2} + \frac{M}{T\eta},$$

which implies

$$\frac{1}{T} \sum_{t=1}^{T} f_t(\theta_t) - \frac{1}{T} \sum_{t=1}^{T} f_t(\theta)$$

$$\leqslant \frac{\eta B}{2} + \frac{M}{T\eta} + \frac{1}{T} \sum_{t=1}^{T} \widehat{f}_t(\theta) - \frac{1}{T} \sum_{t=1}^{T} f_t(\theta)$$

$$\leqslant \frac{\eta B}{2} + \frac{M}{T\eta} + \frac{1}{T} \sum_{t=1}^{T} (\zeta_t - \nabla f_t(\theta_t))^\top (\theta - \theta_t) + \frac{1}{T} \sum_{t=1}^{T} \xi_t^\top (\theta - \theta_t).$$

Now we bound the term $\sum_{t=1}^{T} (\zeta_t - \nabla f_t(\theta_t))^\top (\theta - \theta_t)$. We note the boundedness of the quantities

$$(\zeta_t - \nabla f_t(\theta_t))^\top (\theta - \theta_t) \leqslant \|\zeta_t - \nabla f_t(\theta_t)\| \, 2\sqrt{2M} \leqslant 4B\sqrt{2M}.$$

To control the sum of bounded random variables, we invoke Hoeffding-Azuma inequality, and obtain that for $0 < \delta < 1$,

$$\mathbb{P}\left\{ \frac{1}{T} \sum_{t=1}^{T} (\zeta_t - \nabla f_t(\theta_t))^\top (\theta - \theta_t) \geqslant 8B\sqrt{\frac{M \log(1/\delta)}{T}} \right\} \leqslant \delta.$$

Finally we have $\xi_t^\top (\theta - \theta_t) \leqslant \|\xi_t\| \, 2\sqrt{2M}$. Putting all the pieces together completes the proof. $\quad\square$

### D.6 Proof of Lemma C.3

*Proof.* The gradients of $F$ with respect to $\omega, \theta$ are

$$\nabla_\theta F = (u_\omega + \alpha f_\theta) \nabla_\theta f_\theta,$$
$$\nabla_\omega F = (f_\theta - b - u_\omega) \nabla_\omega u_\omega.$$

First we show for all $x_1, x_2, \omega$ and $\theta$, we have that $\nabla_\theta F$ is bounded. It is easy to see by Lemma 5.1

$$\|\nabla_\theta F\|_2 = \mathcal{O}((1 + \alpha)B).$$

Next we show that for all $\theta, \omega$, $\mathbb{E}_{\text{init},X}[\|\nabla_\theta F_t - \nabla_\theta \widehat{F}\|]$ goes to zero as $m \to \infty$.

$$\mathbb{E}_{\text{init},X}\big[\|\nabla_\theta F - \nabla_\theta \widehat{F}\|\big]$$

$$\leqslant \sqrt{\mathbb{E}_{\text{init},X}\big[\|\nabla_\theta f\|^2\big]\mathbb{E}_{\text{init},X}\big[(u - \hat{u})^2\big]}$$

$$+ \sqrt{\mathbb{E}_{\text{init},X}\big[\|\nabla_\theta \hat{f} - \nabla_\theta f\|^2\big]\mathbb{E}_{\text{init},X}\big[\hat{u}^2\big]}$$

$$+ \alpha\sqrt{\mathbb{E}_{\text{init},X}\big[\|\nabla_\theta f\|^2\big]\mathbb{E}_{\text{init},X}\big[(f - \hat{f})^2\big]}$$

$$+ \alpha\sqrt{\mathbb{E}_{\text{init},X}\big[\|\nabla_\theta \hat{f} - \nabla_\theta f\|^2\big]\mathbb{E}_{\text{init},X}\big[\hat{f}^2\big]}$$

$$= \mathcal{O}((1 + \alpha)B^{3/2}m^{-1/4})$$

$$\square$$

### D.7 Proof of Theorem 4.1

**Remark.** In fact, the two bounds in Theorem 4.1 are also valid bounds on $\mathbb{E}_{\text{init}}\left[\frac{1}{T}\sum_{t=1}^{T} E(f_t)\right]$ and $\frac{1}{T}\sum_{t=1}^{T} E(f_t)$, respectively. For example, in the 2-layer NN case, it also holds that

$$\mathbb{E}_{\text{init}}\left[\frac{1}{T}\sum_{t=1}^{T} E(f_t)\right] = \mathcal{O}\left(a\eta B + \frac{B}{T\eta} + \frac{aB^{3/2}\log^{1/2}(1/\delta)}{T^{1/2}} + \frac{aB^{5/2}}{m^{1/4}}\right). \tag{41}$$

During training we obtain a sequence of NN weights $\theta_1, \theta_2, \cdots, \theta_T$ and the corresponding NNs $f_1, f_2, \ldots, f_T$. The difference lies in that in (41) we bound the *average of the suboptimality* of the NNs $f_1, f_2, \ldots, f_T$ rather than the *suboptimality of the averaged NN* $\bar{f}_T = \frac{1}{T}\sum_t f_t$, as is done in Theorem 4.1. The bound (41) implies that to choose the output NN it suffices to just pick one NN from the sequence of NNs $f_1, f_2, \ldots, f_T$ uniformly.

#### Proof of Theorem 4.1, two-layer NN

*Proof.* Based on the analysis in Appendix C, all we need to do is to estimate the rate of the following quantities

1. $\mathbb{E}_{\text{init}}\left[|\widehat{\phi}(\theta, \omega) - \phi(\theta, \omega)|\right] = \mathcal{O}((1+\alpha)B^{5/2}m^{-1/4})$,

2. $\sup\|\theta\| = \mathcal{O}(B), \|\nabla_\theta F\| = \mathcal{O}((1+\alpha)B)$,

3. $\sup\|\omega\| = \mathcal{O}(B), \|\nabla_\omega F\| = \mathcal{O}(B)$,

4. $\mathbb{E}_{\text{init},X}\left[\|\nabla_\theta F - \nabla_\theta \widehat{F}\|\right] = \mathcal{O}((1+\alpha)B^{3/2}m^{-1/4})$, and

5. $\mathbb{E}_{\text{init},X}\left[\|\nabla_\omega F - \nabla_\omega \widehat{F}\|\right] = \mathcal{O}(B^{3/2}m^{-1/4})$.

The missing pieces are

- $\|\nabla_\omega F\|$ is bounded, and

- $\mathbb{E}_{\text{init},X}\left[\|\nabla_\omega F - \nabla_\omega \widehat{F}\|\right] = \mathcal{O}(B^{3/2}m^{-1/4})$.

First we bound the term $\|\nabla_\omega F\|$. It is easy to see

$$\|\nabla_\omega F\| = \mathcal{O}(B).$$

Then we show $\mathbb{E}_{\text{init},X}\left[\|\nabla_\omega F - \nabla_\omega \widehat{F}\|\right] = \mathcal{O}(B^{3/2}m^{-1/4})$

$$\mathbb{E}_{\text{init},X}\left[\|\nabla_\omega F - \nabla_\omega \widehat{F}\|\right]$$

$$\leqslant \sqrt{\mathbb{E}_{\text{init},X}[|f - b - u|^2]\mathbb{E}_{\text{init},X}[\|\nabla_\omega \hat{u} - \nabla_\omega u\|^2]}$$

$$+ \sqrt{\mathbb{E}_{\text{init},X}[|(f - \hat{f}) + (u - \hat{u})|^2]\mathbb{E}_{\text{init},X}[\|\nabla_\omega \hat{u}\|^2]} \qquad \text{(Cauchy-Schwarz inequality)}$$

$$= \mathcal{O}(B^{3/2}m^{-1/4})$$

$\square$

#### Proof of Theorem 4.1, multi-layer NN

*Proof.* We mimic the same proof technique as the two-layer case. We need to verify with probability at least $1 - \exp(\Omega(\log^2 m))$ over the NN initialization,

1. $|\widehat{\phi}(\theta, \omega) - \phi(\theta, \omega)| = \mathcal{O}((1+\alpha)B^{8/3}H^6 m^{-1/6}\log^{3/2} m)$, for all $\theta, \omega \in S_{B,H}$,

2. $\sup\|\theta\|_2 = H^{1/2}B, \|\nabla_\theta F\| = \mathcal{O}((1+\alpha)B^{4/3}H^4\log m)$ for all $\theta, \omega \in S_{B,H}$ and $x_1, x_2$,

3. $\sup\|\omega\|_2 = H^{1/2}B, \|\nabla_\omega F\| = \mathcal{O}(B^{4/3}H^4\log m)$, for all $\theta, \omega \in S_{B,H}$ and $x_1, x_2$,

4. $\mathbb{E}_X[\|\nabla_\omega F - \nabla_\omega \widehat{F}\|] = \mathcal{O}(B^{4/3} H^4 m^{-1/6} \log^{3/2} m)$, for all $\theta, \omega \in S_{B,H}$, and

5. $\mathbb{E}_X[\|\nabla_\theta F - \nabla_\theta \widehat{F}\|] = \mathcal{O}((1+\alpha) B^{4/3} H^4 m^{-1/6} \log^{3/2} m)$ for all $\theta, \omega \in S_{B,H}$.

To show claim 1, we need to find high probability bounds of the terms

$$|\widehat{u}\widehat{f} - uf|, \tag{42}$$
$$|(\widehat{u} - u)b|, \tag{43}$$
$$|\widehat{u}^2 - u^2| \tag{44}$$
$$|\widehat{f}^2 - f^2| \tag{45}$$

For the term (42),

$$
\begin{aligned}
&|\widehat{u}\widehat{f} - uf| \\
&\leqslant |\widehat{u}(\widehat{f} - f)| + |(\widehat{u} - u)f| \\
&\leqslant \sqrt{\|\widehat{u}\|^2 \|\widehat{f} - f\|^2} + \sqrt{\|f\|^2 \|\widehat{u} - u\|^2} \qquad \text{(Cauchy-Schwarz inequality)} \\
&= \sqrt{\mathcal{O}(B^2 H^3 \log^2 m \cdot B^{8/3} H^6 m^{-1/3} \log m)} \\
&\quad + \sqrt{\mathcal{O}(B^{8/3} H^6 m^{-1/3} \log m) \cdot \mathcal{O}(B^2 H^3)} \tag{46} \\
&= \mathcal{O}(B^{7/3} H^{9/2} m^{-1/6} \log^{3/2} m),
\end{aligned}
$$

where equality (46) is valid with probability at least $1 - \exp(\Omega(\log^2 m))$. Similarly we have the following high probability bounds.

$$
\begin{aligned}
|(\widehat{u} - u)b| &= \mathcal{O}(B^{4/3} m^{-1/6} H^3 \log^{1/2} m), \\
|\widehat{u}^2 - u^2| &= \mathcal{O}(B^{8/3} m^{-1/6} H^6 \log^{3/2} m).
\end{aligned}
$$

Putting all the pieces together completes the proof of claim 1.

For claim 2, $\|W - W(0)\|_2 \leqslant \sqrt{H} B$ implies $\sup \|\theta\|_2 \leqslant H^{1/2} B$. For $\|\nabla_\theta F\|$,

$$
\begin{aligned}
&\|\nabla_\theta F\|_2 \\
&= \|(u + \alpha f)\nabla_\theta f\|_2 \\
&= \mathcal{O}((1+\alpha) B^{4/3} H^4 \log m).
\end{aligned}
$$

This completes proof of claim 2. Claim 3 follows similarly. For claim 4,

$$
\begin{aligned}
&\|\nabla_\omega F - \nabla_\omega \widehat{F}\| \\
&= \|(f - b - u)\nabla_\omega u - (\widehat{f} - b - \widehat{u})\nabla_\omega \widehat{u}\| \\
&\leqslant \sqrt{|\widehat{f} - b - \widehat{u}|^2 \|\nabla_\omega \widehat{u} - \nabla_\omega u\|^2} \\
&\quad + \sqrt{|(f - \widehat{f}) + (u - \widehat{u})|^2 \|\nabla_\omega u\|^2} \\
&= \mathcal{O}(B^{4/3} H^4 m^{-1/6} \log^{3/2} m)
\end{aligned}
$$

where the last equality holds with high probability. Recall the decomposition (22),

$$\frac{1}{T}\sum_{t=1}^{T} \phi_t(\theta_t) - \frac{1}{T}\sum_{t=1}^{T} \phi_t(\theta) \tag{22, revisited}$$

$$= \underbrace{\frac{1}{T}\sum_{t=1}^{T} \phi_t(\theta_t) - \frac{1}{T}\sum_{t=1}^{T} \widehat{\phi}_t(\theta_t)}_{(19)} + \underbrace{\frac{1}{T}\sum_{t=1}^{T} \widehat{\phi}_t(\theta_t) - \frac{1}{T}\sum_{t=1}^{T} \widehat{\phi}_t(\theta)}_{(20)} + \underbrace{\frac{1}{T}\sum_{t=1}^{T} \widehat{\phi}_t(\theta) - \frac{1}{T}\sum_{t=1}^{T} \phi_t(\theta)}_{(21)}.$$

Finally, we put together the pieces. Define the events

$$E_1 = \left\{ \underbrace{\frac{1}{T}\sum_{t=1}^{T}\phi_t(\theta_t) - \frac{1}{T}\sum_{t=1}^{T}\widehat{\phi}_t(\theta_t)}_{(19)} + \underbrace{\frac{1}{T}\sum_{t=1}^{T}\widehat{\phi}_t(\theta) - \frac{1}{T}\sum_{t=1}^{T}\phi_t(\theta)}_{(21)} \right.$$

$$\left. = \mathcal{O}((1+\alpha)B^{8/3}H^6 m^{-1/6}\log^{3/2}m) \right\}$$

and

$$E_2 = \left\{ \underbrace{\frac{1}{T}\sum_{t=1}^{T}\widehat{\phi}_t(\theta_t) - \frac{1}{T}\sum_{t=1}^{T}\widehat{\phi}_t(\theta)}_{(20)} \right.$$

$$\left. = \mathcal{O}(P_1\eta a\log m + \frac{P_2}{T\eta} + \frac{aP_3\log m\log^{1/2}(1/\delta)}{T^{1/2}} + \frac{P_3 a\log^{3/2}m}{m^{1/6}}) \right\}$$

where $P_1 = H^4 B^{4/3}$, $P_2 = H^{1/2}B$ and $P_3 = H^5 B^2$, defined in Theorem 4.1. By claim 1 we have $\mathbb{P}(E_1) \geqslant 1 - \exp(\Omega(\log^2 m))$. By claim 2, claim 5 and Lemma C.2 we have $\mathbb{P}(E_2) \geqslant 1 - \delta - \exp(\Omega(\log^2 m))$. Then $\mathbb{P}(E_1 \cap E_2)) \geqslant 1 - c\delta - c\exp(\Omega(\log^2 m))$ for some absolute constant $c$. The same analysis applies for $\omega$ and therefore we complete the proof. $\square$

## D.8 Proof of Theorem 4.2

The proof relies on the following lemma that controls the regularization bias by imposing smoothness assumption on the truth.

**Lemma D.3** (Hilbert scale and regularization bias)**.** *Assume the operator $A$ in* (1) *is injective and compact. Let $f^\alpha = \operatorname{argmin}_{f \in \mathcal{H}} \frac{1}{2}\|Af - b\|_{\mathcal{E}}^2 + \frac{\alpha}{2}\|f\|_{\mathcal{H}}^2$ for some $\alpha > 0$. If the solution $f$ to* (1) *lies in the regularity space $\Phi_\beta$ defined in* (15) *for some $\beta > 0$, then*

$$\|f - f^\alpha\|_{\mathcal{H}}^2 = O(\alpha^{\min\{\beta,2\}}).$$

*Proof.* See Section 3.3 of [11]. $\square$

Compactness of a conditional expectation operator is a mild condition; see Appendix E for a discussion.

We remark that four quantities are involved in this proof: the truth $f$ that uniquely solves $Af = b$, the Tikhonov regularized solution $f^\alpha$ defined in (5), the Tikhonov regularized solution approximated by the class of NNs (see Equation (7)), denoted $f_{\mathrm{NN}}^\alpha$, and the average of the iterates generated by Algorithm 1, $\bar{f}_T$. Lemma D.3 provides a bound on the gap between $f$ and $f^\alpha$; Theorem 4.1 controls $f_{\mathrm{NN}}^\alpha - \bar{f}_T$. Theorem 4.2 assumes that $f_{\mathrm{NN}}^\alpha = f^\alpha$. See Section G for a graphical representation.

We start with the decomposition of $\|\bar{f}_T - f\|_{\mathcal{H}}^2$

$$\|\bar{f}_T - f\|_{\mathcal{H}}^2 \leqslant 2\|\bar{f}_T - f^\alpha\|_{\mathcal{H}}^2 + 2\|f^\alpha - f\|_{\mathcal{H}}^2.$$

Here the first term on the RHS represents optimization error and the second term is regularization bias. Lemma D.3 provides a bound on the second term. Now we bound the first term.

Recall the definition of Tikhonov regularized functional for a compact linear operator $A$

$$L(f) = L_\alpha(f) = \frac{1}{2}\|Ag - b\|_{\mathcal{E}}^2 + \frac{\alpha}{2}\|f\|_{\mathcal{H}}^2.$$

Denote by $f^\alpha$ the unique minimizer of $L$ over $\mathcal{H}$. This is always well-defined for a compact linear operator $A$. We want to show the strong convexity of $L_\alpha$, i.e.,

$$\frac{\alpha}{2}\|\bar{f}_T - f^\alpha\|_{\mathcal{H}}^2 \leqslant L_\alpha(\bar{f}_T) - L_\alpha(f^\alpha). \tag{47}$$

If (47) is true, under the conditions of Theorem 4.1 (2-layer NN case), we have with probability at least $1 - \delta$ over the sampling process,

$$\mathbb{E}_{\text{init}}[\|\bar{f}_T - f^\alpha\|_{\mathcal{H}}^2] \leqslant \frac{2}{\alpha} \mathbb{E}_{\text{init}}[L_\alpha(\bar{f}_T) - L_\alpha(f^\alpha)] \tag{48}$$

$$= \frac{2}{\alpha} \mathcal{O}\Big(a\eta B + \frac{B}{T\eta} + \frac{aB^{3/2} \log^{1/2}(1/\delta)}{T^{1/2}} + \frac{aB^{5/2}}{m^{1/4}}\Big). \tag{49}$$

Setting $\eta = (aT)^{-1/2}$ where $a = \max\{\alpha, 1\}$, and combining results from Lemma D.3 and (49) we complete the proof.

Now we show (47). For all $x \in \mathcal{H}$, $x + h \in \mathcal{H}$,

$$2L_\alpha(x + h) = \|A(x + h) - b\|_{\mathcal{E}}^2 + \alpha\|x + h\|_{\mathcal{H}}^2 \tag{50}$$

$$= \|Ax - b\|_{\mathcal{E}}^2 + \|Ah\|_{\mathcal{E}}^2 + \langle Ax - b, Ah \rangle_{\mathcal{E}} + \alpha\|x\|_{\mathcal{H}}^2 + \alpha\|h\|_{\mathcal{H}}^2 + 2\alpha\langle x, h \rangle_{\mathcal{H}} \tag{51}$$

$$= 2L_\alpha(x) + 2\alpha\langle x, h \rangle_{\mathcal{H}} + 2\langle Ax - b, Ah \rangle_{\mathcal{E}} + \|Ah\|_{\mathcal{E}}^2 + \alpha\|h\|_{\mathcal{H}}^2 \tag{52}$$

$$= 2L_\alpha(x) + 2\alpha\langle x, h \rangle_{\mathcal{H}} + 2\langle A^*(Ax - b), h \rangle_{\mathcal{H}} + \|Ah\|_{\mathcal{E}}^2 + \alpha\|h\|_{\mathcal{H}}^2 \tag{53}$$

$$= 2L_\alpha(x) + 2\langle \alpha x + A^*Ax - A^*b, h \rangle_{\mathcal{H}} + \|Ah\|_{\mathcal{E}}^2 + \alpha\|h\|_{\mathcal{H}}^2. \tag{54}$$

Moreover, the regularized solution $f^\alpha$ is given by the unique solution to the equation $\alpha f^\alpha + A^*Af^\alpha = A^*b$ and depends continuously on $b$ [30]. Setting $x = f^\alpha$, $h = f - f^\alpha$ and applying $\alpha f^\alpha + A^*Af^\alpha = A^*b$ complete the proof of (47).

## E   Compactness of conditional expectation operators

Let $X = [X_1^\top, X_2^\top]^\top$ be a random vector with distribution $F_X$ and let $F_{X_1}, F_{X_2}$ be the marginal distributions of $X$ and $Y$, respectively. Assume there is no common elements in $X_1$ and $X_2$. Define Hilbert spaces $\mathcal{H} = L^2(X_1)$ and $\mathcal{E} = L^2(X_2)$. Let $A$ be the conditional expectation operator:

$$A : \mathcal{H} \to \mathcal{E}$$
$$f(\cdot) \to \mathbb{E}[f(X_1) \mid X_2 = \cdot].$$

If there is no common elements in $X_1$ and $X_2$, compactness of an conditional expectation operator is in fact a mild condition [11]. If p.d.f.s of $X, X_1$ and $X_2$ exist, denoted $f_X, f_{X_1}$ and $f_{X_2}$, then $A$ can be represented as an integral operator with kernel

$$k(x_1, x_2) = \frac{f_{X_1, X_2}(x_1, x_2)}{f_{X_1}(x_1) f_{X_2}(x_2)},$$

and $(Af)(x_2) = \int k(x_1, x_2) f(x_1) f_{X_1}(x_1) dx_2$. In this case, a sufficient condition for compactness of $A$ is

$$\iint \left[ \frac{f_{X_1, X_2}(x_1, x_2)}{f_{X_1}(x_1) f_{X_2}(x_2)} \right]^2 f_{X_1}(x_1) f_{X_2}(x_2) dx_1 dx_2 < \infty.$$

We now discuss well-posedness of (1). The operator equation (1) is called well-posed (in Hadamard's sense) if (i) *(existence)* a solution $f$ exists, (ii) *(uniqueness)* the solution $f$ is unique, and (iii) *(stability)* the solution $f$ is continuous as a function of $b$. More precisely, if $A : \mathcal{H} \to \mathcal{E}$ is bijective and the inverse operator $A^{-1}$ is continuous, then equation (1) is well-posed [30]. Injectivity is usually a property of the data distribution and is tantamount to assuming identifiability of the structural function

## F   A comment on Dual IV

In this section, we review the work of Dual IV [36] and point out the differences between their work and ours. Dual IV considers nonparametric IV estimation using min-max game formulation and bears similarities with this work. However, we remark that our framework (1) includes a wide range of models, including IV regression, and that the use of NNs and detailed analysis on the convergence of the training algorithm also distinguish our work from Dual IV. The goal of this section is to show the resulting min-max problem for IV regression in this paper has a natural connection with GMM.

Recall that IV regression considers the following conditional moment equation

$$\mathbb{E}[Y - g(X) \mid Z] = 0. \tag{2, revisited}$$

Let $\mathcal{G}$ be an arbitrary class of continuous functions which we assume contains the truth that fulfills the integral equation. Dual IV proposes to solve

$$\min_{g \in \mathcal{G}} R(g) := \mathbb{E}_{YZ}\left[\left(Y - \mathbb{E}[g(X) \mid Z]\right)^2\right], \tag{55}$$

while this paper solves

$$\min_{g \in \mathcal{G}} L(g) = \|Af - b\|_{\mathcal{E}}^2 = \mathbb{E}_Z\left[\left(\mathbb{E}[Y \mid Z] - \mathbb{E}[g(X) \mid Z]\right)^2\right],$$

an unregularized version of Example 1. The operator $A$ and $b \in \mathcal{E}$ are defined in Example 1.

To introduce the maximizer, Dual IV [36] resorts to the interchangeability principle.

**Lemma F.1** (Interchangeable principle). *Let $(\Omega, \mathcal{F}, \mathbb{P})$ be a probability space, $f : \mathbb{R}^n \times \Omega \to \mathbb{R} \cup \{+\infty\}$, and $\mathcal{L}_2 = \mathcal{L}_2(\Omega, \mathcal{F}, \mathbb{P})$ be the class of square integrable functions. Let $\mathcal{X}$ be the set of mappings $\chi : \Omega \to \mathbb{R}^n$ such that $f_\chi \in \mathcal{L}_2$, where $f_\chi(\cdot) := f(\chi(\cdot), \cdot)$. Assume $F(\omega) := \sup_{x \in X} f(x, \omega) \in \mathcal{L}_2$ and that $f$ is upper semi-continuous[2]. Then the following holds.*

$$\mathbb{E}\left[\sup_{x \in X} f(x, \omega)\right] = \sup_{\chi \in \mathcal{X}} \mathbb{E}[f(\chi(\omega), \omega)].$$

*Proof.* See Proposition 2.1 in [43]. See also Proposition 1 in [16] for a proof for the case where $f : \mathbb{R} \times \Omega \to \mathbb{R}$. $\qquad\square$

With the interchangeability principle, (55) can be rewritten as

$$\min_{g \in \mathcal{G}} \max_{u \in \mathcal{U}} \Psi(g, u) := \mathbb{E}_{XYZ}[(g(X) - Y)u(Y, Z)] - \tfrac{1}{2}\mathbb{E}_{YZ}\left[u(Y, Z)^2\right].$$

By comparison, an unregularized version of the min-max problem derived in this paper (23) is

$$\min_{g \in L^2(X)} \max_{u \in L^2(Z)} \mathbb{E}_{XYZ}[(g(X) - Y) \cdot u(Z) - \tfrac{1}{2}u^2(Z)]. \tag{56}$$

The absence of the variable $Y$ in the maximizer $u$ in (56) facilitates a natural connection between (56) and GMM.

To achieve such interpretation, we first introduce a GMM estimator for (2). The conditional moment restriction (2) implies that for a set of functions $f_1, f_2, \ldots, f_m$ of $Z$, we have $\mathbb{E}\left[(Y - g(X))f_j(Z)\right] = 0$. Define by $\psi(f, g) := \mathbb{E}_{XYZ}[(Y - g(X))f(Z)]$ the moment violation function, and the GMM estimator

$$g_{\text{GMM}} \in \arg\min_{g \in \mathcal{G}} \frac{1}{2} \sum_{j=1}^m \psi(f_j, g)^2.$$

Collect the moment violations by a vector $\psi_v(g) := (\psi(f_1, g), \ldots, \psi(f_m, g))^\top \in \mathbb{R}^m$. To achieve efficiency the moments are usually weighted. Let $\Lambda$ be a $m$ by $m$ symmetric matrix. We define the quadratic norm $\|\phi\|_\Lambda^2 = \phi^\top \Lambda \phi$ given a vector $\phi$.

Now we are ready to state the connection between GMM and (56). Define the space of maximizer $\mathcal{U} = \text{span}\{f_1, \ldots, f_m\}$. We focus on the inner maximization of (56). Define

$$J(g) := \max_{u \in \mathcal{U}} \mathbb{E}_{XYZ}[(g(X) - Y) \cdot u(Z) - \tfrac{1}{2}u^2(Z)].$$

Note that maximizer is now constrained in $\mathcal{U}$. Mimicking Theorem 5 in [36], we can show $J(g)$ is in fact a weighted sum of the moment violations $\{\psi(f_j, g)\}$.

**Lemma F.2.** *Let $f_1, f_2, \ldots, f_m$ be a set of real-valued functions of $Z$. Define the weight matrix $\Lambda := \mathbb{E}_Z[\mathbf{f}(Z)\mathbf{f}(Z)^\top]$ where $\mathbf{f} := (f_1(Z), \ldots, f_m(Z))^\top$. Then $J(g) = \tfrac{1}{2}\|\psi_v(g)\|_{\Lambda^{-1}}^2$.*

*Proof.* The proof is identical to Appendix C of [36] except for replacing $f(Y, Z)$ with $f(Z)$. The proof relies on simple algebra manipulation and is presented for completeness. For any $u \in \mathcal{U}, u = \sum_{j=1}^{m} \alpha_j f_j$ for some $\boldsymbol{\alpha} = (\alpha_1, \dots, \alpha_m)^\top \in \mathbb{R}^m$.

$$
\begin{aligned}
J(g) &= \max_{\alpha \in \mathbb{R}^m} \mathbb{E}_{XYZ} \left[ (g(X) - Y) \left( \sum_{j=1}^{m} \alpha_j f_j(Z) \right) \right] - \frac{1}{2} \mathbb{E}_Z \left[ \left( \sum_{j=1}^{m} \alpha_j f_j(Z) \right)^2 \right] \\
&= \max_{\boldsymbol{\alpha} \in \mathbb{R}^m} \sum_{i=1}^{m} \alpha_j \mathbb{E}_{XYZ} \left[ (g(X) - Y) f_j(Z) \right] - \frac{1}{2} \mathbb{E}_Z \left[ \left( \sum_{j=1}^{m} \alpha_j f_j(Z) \right)^2 \right] \\
&= \max_{\boldsymbol{\alpha} \in \mathbb{R}^m} \boldsymbol{\alpha}^\top \psi_v - \frac{1}{2} \boldsymbol{\alpha}^\top \Lambda \boldsymbol{\alpha} \\
&= \frac{1}{2} \psi_v^\top \Lambda^{-1} \psi_v.
\end{aligned}
$$

$\square$

Lemma F.2 shows that if the maximizer is constrained to be in the span of a set of pre-defined test functions $\{f_j\}$, the minimization in (56) in fact produces a weighed GMM estimator. In contrast, the GMM interpretation provided in Section 3.5 of [36] requires the definition of a so-called augmented IV $W := (Y, Z)$. It is unnatural to view the response variable $Y$ as a component of the IV.

# G A roadmap to the proof of Theorem 4.2

In Figure 3 we can see throughout the discussion we make a couple of simplifying assumptions (e.g., Assumption A.5 assumes the conditional expectation operator is close in $\mathcal{F}_{\text{NN}}$, and Assumption A.6 assumes the primal problems (7) and (5) give the same solution). These assumptions are justified by the representation power of NNs. One could instead explicitly incorporate approximation error in the bounds.

Figure 3: Relation between the quantities of interest. Texts above/near the arrows summarize the key elements of connecting different problems.

## Footnotes

[2]Random upper semi-continuous, to be precise.