[Reviews · NeurIPS 2020]

Review 1

Summary and Contributions: The paper proposes an adversarial minimax two player game approach for optimising the parameters of a generalised structural equation model (SEM) formulated as a saddle-point problem. The generalised SEM is defined in terms of a conditional expectation operator mapping between a hilbert space of structural functions of interest to a hilbert space of known or estimated functions of the outcome. These spaces are subsequently chosen to be the space of possible neural networks and a stochastic primal-dual algorithm is given for finding a solution to the saddle-point problem. Furthermore, the work proves global convergence of the algorithm. This main result is achieved, under certain specific data and weight initialisation conditions, using a regret analysis while considering the infinite width limit for neural networks that cause them to behave like linear learners.

Strengths: *A tractable estimation algorithm for generalised SEMs. *Derived results for precise statements about the rate of error in estimation. *A proof of global convergence of the algorithm.

Weaknesses: *No empirical validation or performance analysis is given for the prosed algorithm, although I understand that the point of the paper is perhaps solely as a theoretical contribution. *The construction of the proposed algorithm seems to largely overlaps with prior work (Muandet et al., 2019), however, there remain significant differences between the two papers and the convergence proof is unique to this work. -- K. Muandet, A. Mehrjou, S. K. Lee, and A. Raj. Dual IV: A single stage instrumental variable regression. arXiv preprint arXiv:1910.12358, 2019.

Correctness: To best of my knowledge, the claims in the paper seem correct, however I did not check all the proofs provided in the appendix. Below are minor suggestions: ### minor ### *Line 48 (.. works in incorporating ..): remove "in" *Line 194/195 (b_1, ..., b_r): Should the subscript not index from 1 to m?

Clarity: The paper is well written and the presentation of the work is also well structured, following a sensible progression.

Relation to Prior Work: The paper includes a related work section which is adequate, with additional discussion on specific connections given in the appendix. In particular, a specific section is dedicated to the work on Dual IV (Muandet et al., 2019), where that work is compared and contrasted to the presented work.

Reproducibility: Yes

Additional Feedback: I would like to complement the authors on including so many detailed examples of SEMs, which I thought was a nice addition to the paper. Although as mentioned above, I can understand that the work is primarily focused on being a theoretical contribution, however, for applied practitioners of SEM models, the paper might have more appeal if the algorithm is demonstrated and compared to other approaches on a real-world practical problem, e.g. the demand estimation problem considered in the Dual IV paper as well as in other prior work. ### Post-rebuttal update ### I thank the authors for their response and appreciate their willingness to include an experimental section in their work (which will only strengthen the presentation) as well as highlighting the comparison to Dual IV. I remain with my original evaluation, seeing this as a good submission.


Review 2

Summary and Contributions: This paper proposes a new estimation procedure of structural equation models based on a min-max game formulation in which both players are represented by neural networks. The authors show that the algorithm they derive converges and is consistent in the sense that the estimate obtained is close to the solution of a regularized version of the orignal structural equation problem.

Strengths: This study introduces a complete framework for generalized structural equation models in which the original problem is first regularized prior to being reformulated (through a saddle-point reformulation) using neural networks to approximate the functions to be optimized. The theoretical guarantees provided in the study further show that the solution provided is of "good quality" (as usual when the number of samples is sufficiently large).

Weaknesses: The main weakness to my opinion lies in the lack of empirical validation of the proposal. I would have liked to see how the algorithm behaves in practice on different datasets for which the number of samples is limited.

Correctness: The claims are correct as far as I can tell (the paper is purely theoretical; no experiments have been conducted).

Clarity: The paper is definitely well written.

Relation to Prior Work: As far as I can tell, the authors have clearly positioned their work with respect to previous contributions. The novelty is clear.

Reproducibility: Yes

Additional Feedback: Two minor remarks: 1. The notation E_{init} should be explained. 2. Theorem 5.2, Eq. 15, replace f with f^{\alpha}.


Review 3

Summary and Contributions: The paper proposes a framework to solve structural equation models formalized as a class of inverse problems where both the the data and the operator can be stochastic. In particular, the authors propose and analyze an approach based on the solution of saddle point problem over classes of neural networks. The statistical properties of the proposed method are analyzed.

Strengths: The study is thorough and complete. The mathematical analysis seems sound. The problem relevant.

Weaknesses: I miss some clearer explanation of why the Tikhonov approach is problematic and the saddle point approach with neural network needed. What are the precise assumption on the inverse problem under study? Some basic experiments are missing.

Correctness: The analyses appears to be sound and correct.

Clarity: It's fairly well written but I got lost in some key points.

Relation to Prior Work: Quite good, but I feel some more reference to the huge literature in inverse problems should be added placing the contribution in that context.

Reproducibility: Yes

Additional Feedback: I found the response satisfying. I encourage the authors to improve the presentation .

[Author Response · NeurIPS 2020]

We appreciate the valuable comments from the reviewers, which will lead to a largely improved final paper. Following
the unanimous suggestion by three reviewers, we will provide a simulation section in the final version.

**Reviewer 2.** We are thrilled to find that Reviewer 2 appreciates our goal to provide a theoretical foundation for the use
of NNs in econometrics models and causal inference where conditional moment equations play a key role in identifying
the structural parameters. We further point to Babii and Florens [1], which includes a battery of economics models that
involve conditional moment restrictions, demonstrating the potential application of our method in those models.

Comparison with Dual IV We thank R2 for noticing that we provided a detailed comparison with Dual IV in Appendix
F. The main difference is that we use a variational characterization of conditional expectation while dual IV resort to
maximality principle (Lemma F.1). Our derivation has a natural connection to generalized method of moments.

**Reviewer 4.** The notation $\mathbb{E}_{\text{init}}[\cdot]$ It is the expectation taken over the random variables $\Xi_0$ or $\Xi_{H,0}$, defined in Eq. (9)
and (10), respectively. We will define it explicitly in the final draft.

The bound of Theorem 4.2 As for the second comment, we emphasize that it should be $f$ that appears in (15), the
solution to Eq. (1), not $f^\alpha$. We will list separately several assumptions made in Theorem 4.2 so that it is easier to parse.

**Reviewer 5.** The need for saddle-point formulation with NNs For example, consider the nonparametric IV problem
$\mathbb{E}[g(X)|Z] = \mathbb{E}[Y|Z]$ where we want to solve for $g$. Consider the square loss $L(g) = \mathbb{E}_Z\left[\left(\mathbb{E}[g(X) - Y|Z]\right)^2\right]$ without
Tikhonov penalty. Assume $g$ is approximated by a function parameterized with $\theta$. Taking the gradient w.r.t. $\theta$ and
assuming exchange of $\nabla_\theta$ and $\mathbb{E}$, we get $\nabla_\theta L(g) = 2\mathbb{E}_Z\left[\mathbb{E}[g(X) - Y|Z] \cdot \mathbb{E}[\nabla_\theta g(X)|Z]\right]$. Assume we observe iid
samples of $(X, Y, Z)$. The product of two expectation terms implies that, to obtain an unbiased estimate of the gradient,
we will need two samples of $(X, Y, Z)$ with $Z$ taking the same value. This is usually unlikely except for simulated
environments.

We can see saddle-point formulation eliminates the need of double samples. The double-sample challenge is a
consequence of the nested structure of the problem, where the conditional expectation operator appears inside the
square. This point was briefly mentioned in the beginning of Sec 2.2 and will be elaborated on in the final draft. The
double-sample problem is also a typical issue in reinforcement learning literature. The "double sampling" problem
becomes more problematic if we use other convex losses. We use NNs as approximators due to their representation
power.

Assumptions on the inverse problem under study We study an instance of inverse problem where the operator is a
conditional expectation operator. We assume (i) the equation $Af = b$ admits a unique solution $f$, and (ii) $A$ is compact
and $f$ is "smooth" w.r.t to $A$. Under (i) we are working with a well-specified and identified model. Under (ii) we have a
bound on the regularization bias (Lemma D.2). A sufficient condition for compactness of $A$ is given in Appendix E.
These assumptions are common in condition moment equation problems [1]. We also emphasize that the identification
problem is very important and well studied in the literature, but we do not focus on this.

Relation to literature of inverse problems We focus on a very specific type of inverse problem. We contrast our method
with two lines of work in conditional moment equations (CME) estimation: (i) non-parametric approach, and (ii)
recent use of NN in CME. This is discussed in the Related Work section. We will provide additional discussion on the
relationship of our method to other methods for solving inverse problems.

In stochastic inverse problem where the operator must be estimated from the data, nonparametric methods proceed
in a two-stage type manner. First estimate the conditional expectation operator, using either sieve estimator or kernel
estimator, and then solve a Tikhonov penalized problem [5]. However, these methods have serious drawbacks: (1)
their performance depends on the choice of kernel function or spline basis; and (2) they become computationally
intractable in high-dimensional feature spaces or with large numbers of training examples. The disadvantages of
nonparametric methods have motivated recent works incorporating NN in CME, especially in instrumental variable
regression. However, these methods suffer either the need of double sampling [3,4], unknown convergence behavior
[1,4], or computational burden [3]. Our method is scalable, provably efficient and does not require double sampling.

**References**

[1] A. Babii and J.-P. Florens. *Is completeness necessary? Estimation in nonidentified linear models*, page 5, footnote 4.

[2] K. Muandet, A. Mehrjou, S. K. Lee, and A. Raj. Dual IV: A single stage instrumental variable regression.

[3] J. Hartford, G. Lewis, K. Leyton-Brown, and M. Taddy. Deep IV: A flexible approach for counterfactual prediction.

[4] G. Lewis and V. Syrgkanis. Adversarial generalized method of moments.

[5] M. Carrasco, J.-P. Florens, and E. Renault. Linear inverse problems in structural econometrics estimation based on spectral
decomposition and regularization. Handbook of Econometrics.


[Meta-Review · NeurIPS 2020]

The paper was reviewed by experts on the topic and discussed after authors rebuttal. Results were found to be interesting and valuable. The reviewers comments should be taken into account while preparing the final version of the paper.